# Phosphomonoesterase and phosphodiesterase activities in the eastern Mediterranean in two contrasting seasonal situations

**France Van Wambeke**[1]**, Pascal Conan**[2,3]**, Mireille Pujo-Pay**[2]**, Vincent Taillandier**[4]**, Olivier Crispi**[2]**, Alexandra Pavlidou**[5]**, Morgane Didry**[1]**, Christophe Salmeron**[3]**, and Elvira Pulido-Villena**[1]

[1]Aix-Marseille Université, Université de Toulon, CNRS , IRD, Mediterranean Institute of Oceanography (MIO), Marseille, France

[2]Sorbonne Université, CNRS, Laboratoire d'Océanographie Microbienne (LOMIC), Observatoire Océanologique, Banyuls-sur-Mer, France

[3]Sorbonne Université, CNRS OSU STAMAR, 4 Place Jussieu, Paris, France

[4]Sorbonne Université, CNRS, Laboratoire d'Océanographie de Villefranche (LOV), Villefranche-sur-Mer, France

[5]Hellenic Centre for Marine Research (HCMR), Institute of Oceanography, 46.7 km Athens-Sounio Av., 19013, Anavyssos, Attica, Greece

**Correspondence:** France Van Wambeke (france.van-wambeke@mio.osupytheas.fr)

**Abstract.** Hydrolysis of dissolved organic phosphorus by marine planktonic microorganisms is a key process in the P cycle, particularly in P-depleted, oligotrophic environments. The present study assessed spatiotemporal variations in phosphomonoesterase (PME) and phosphodiesterase (PDE) activities using concentration kinetics in the eastern Mediterranean Sea in two contrasting situations: the end of winter (including a small bloom period) and autumn. The distribution and regulation of the maximum hydrolysis rate (Vm) and half-saturation constant (Km) of both ectoenzymes were assessed in relation to the vertical structure of the epipelagic layers. PME reached its maximum activities (Vm) after the addition of 1 µM MUF-P (4-methylumbelliferyl phosphate), whereas, for PDE, it was necessary to add up to 50 µM bis(4-methylumbelliferyl)phosphate (bis-MUF-P) to reach saturation state. On average, the Km of PDE was $33 \pm 25$ times higher than that of PME. The Vm of PME and Vm of PDE were linearly correlated. Conversely to the Km values, Vm values were on the same order of magnitude for both ectoenzymes, with their ratio (Vm PME : Vm PDE) ranging between 0.2 and 6.3. Dissolved organic phosphorus (DOP) and the phosphomonoesterase hydrolysable fraction of DOP explained most of the lack of variability in Vm PME and Vm PDE. On the contrary, Vm of both phosphohydrolase enzymes was inversely correlated to the concentration of dissolved inorganic phosphorus. The particular characteristics of concentration kinetics obtained for PDE (saturation at 50 µM, high Km, high turnover times) are discussed with respect to the possible unequal distribution of PDE and PME among the size continuum of organic material and accessibility of phosphodiesters.

## 1 Introduction

Some species of phytoplankton and heterotrophic prokaryotes (Hprok) have the genetic ability to produce ectoenzymatic phosphatases that provide an important alternative source of P through extracellular hydrolysis of dissolved organic phosphorus (DOP). DOP is composed of various compounds having different degrees of bioavailability (Karl, 2014) including phosphate mono- and diesters (Kolowith et al., 2001; Yamaguchi et al., 2019). Determining ectoenzymatic activity using artificial fluorogenic substrates like 4-methylumbelliferyl phosphate is relatively simple and sensitive (Hoppe, 1983), and this method has been widely used in all oceanic regions to measure phosphomonoesterase (PME) activity (Su et al., 2023). Over large spatiotemporal scales, PME activity is usually regulated by dissolved inorganic phosphorus (DIP) with an increase in the activity when the

DIP concentration decreases. High rates of PME have been encountered in well-known P-limited environments like the Mediterranean Sea or the Sargasso Sea (Van Wambeke et al., 2002; Lomas et al., 2010; Pulido-Villena et al., 2021). Thus, PME activities have been extensively used as indicators of P deficiency (Sala et al., 2001; Van Wambeke et al., 2002; Labry et al., 2005; Lomas et al., 2010; Zaccone et al., 2012). However this inverse relationship between DIP concentration and PME activity is not systematic (Hoppe and Ullrich, 1999; Labry et al., 2016; Davis and Mahaffey, 2017; Duhamel et al., 2021; Lidbury et al., 2022), and questions are raised about, for instance, the role played by the consortium of microorganisms present, the genetic nature of the phosphatase produced, its localization (i.e., periplasmic or truly dissolved), its dependence on or independence of metal ions, and its promiscuity (Luo et al., 2009; Baltar et al., 2010; Mahaffey et al., 2014; Cerdan-Garcia et al., 2021; Srivastava et al., 2021; Lidbury et al., 2022). Kinetic parameters of PME have been also assayed using multiple concentrations of the substrate, providing additional information on the maximum rate (Vm), half-saturation constant (Km) and turnover times (Km : Vm ratio) with respect to environmental conditions (Labry et al., 2005; Duhamel et al., 2011; Suzumura et al., 2012; Pulido-Villena et al., 2021). The enzymatically hydrolysable fraction of DOP (here labile DOP, $L_{DOP}$) is defined as the DOP fraction hydrolyzed by a commercially available alkaline phosphatase, under optimal conditions of enzyme concentration, pH and temperature (Feuillade and Dorioz, 1992). The dynamics of both DIP and $L_{DOP}$ have been investigated in the western North Pacific (Hashihama et al., 2013), as well as the central North Pacific (Yamaguchi et al., 2019), demonstrating the importance of $L_{DOP}$ in supporting productivity in oligotrophic regions.

In addition to phosphomonoesters, phosphodiesters (P-diesters) also constitute an enzymatically hydrolysable pool in DOP. In aquatic environments, typical P-diesters identified are nucleotides, nucleic acids and phospholipids coming from microorganism's intracellular material (Karl and Björkman, 2015), but the methodology used to estimate the P-diester pool (using also a commercially purified phosphodiesterase enzyme; Monbet et al., 2007; Yamaguchi et al., 2019) does not allow for determining the in situ P-diester chemical composition in detail. Phosphodiesterase (PDE) activity is also determined using artificial substrates like bis-4-methylumbelliferyl phosphate, bis-paranitrophenyl phosphate or paranitrophenyl thymidine 5'-monophosphate. PDE activity has been detected in cultures of marine heterotrophic bacteria (Dunlap and Callahan, 1993; Noskova et al., 2019), in a dinoflagellate culture causing harmful bloom (Huang et al., 2021) and in a diatom culture (Yamaguchi et al., 2013). After enrichment of various chemical forms of organic phosphate added experimentally, the large changes in taxonomic diversity and activity of heterotrophic bacteria and phytoplankton (Muscarella et al., 2014; Sisma-Ventura and Rahav, 2019; Filella et al., 2022), as well as expression of different phosphatase genes (Zheng et al., 2019), suggest an important role of the DOP availability in shaping microbial diversity in aquatic environments. Nevertheless, PDE has been already measured in environmental conditions in eutrophic aquatic systems (Jørgensen et al., 2015) or coastal areas (Huang et al., 2022), with few studies describing PDE activity in oceanic waters and only in the Pacific Ocean (Sato et al., 2013; Yamaguchi et al., 2019; Thomson et al., 2020; Srivastava et al., 2021).

The eastern Mediterranean Sea is particularly impoverished in P relative to N, leading to high N : P molar ratios (Durrieu De Madron et al., 2011; Powley et al., 2017). The depth gap separating the two nutriclines increases eastward as the phosphacline deepens faster than the nitracline (Pujo-Pay et al., 2011). Surface concentrations of DIP are typically under 50 nM (Djaoudi et al., 2018b), whereas nitrate is present at the surface after winter convection events which are strong enough to reach the nitracline (Ben-Ezra et al., 2021; D'Ortenzio et al., 2021). Through in situ enrichment experiments, in minicosms or in bioassays, it has been shown that primary producers and heterotrophic prokaryotes within the surface layers of the eastern Mediterranean Sea are primarily limited by P, although this is sometimes accompanied by a co-limitation with N for phytoplankton and N or labile C for heterotrophic prokaryotes (Zohary and Robarts, 1998; Van Wambeke et al., 2002; Thingstad et al., 2005; Tanaka et al., 2011; Sisma-Ventura and Rahav, 2019). Consequently, P availability plays a major role in the microbial food web functioning in the eastern Mediterranean Sea.

We propose here an analysis of the concentration kinetic parameters, including the maximum rates, half-saturation constant and turnover time, of the two types of phosphoesterases, PME and PDE, in the eastern Mediterranean around Crete during two distinct seasons: in autumn (October), chosen to represent typical warm and strong oligotrophic conditions, and in late winter (February–March), chosen to illustrate the productive conditions associated with episodic phytoplankton blooms. Our analysis aims to characterize the distribution of PME and PDE activities in connection with the distribution of DOP, $L_{DOP}$ and DIP in the epipelagic layers in this area, which is recognized as one of the most P-limited marine environments. A second paper in preparation (Van Wambeke et al., 2024) will be dedicated to PDE and PME distribution within the surface mixed layer in relation to mesoscale variability (cyclones vs. anticyclones) and the progression of the phytoplankton bloom in winter.

## 2 Material and methods

### 2.1 Sampling

Two cruises were conducted during the period 2018–2019: PERLE1 (11–20 October 2018) and PERLE2 (27 February–15 March 2019). These cruises were the basis for an exten-

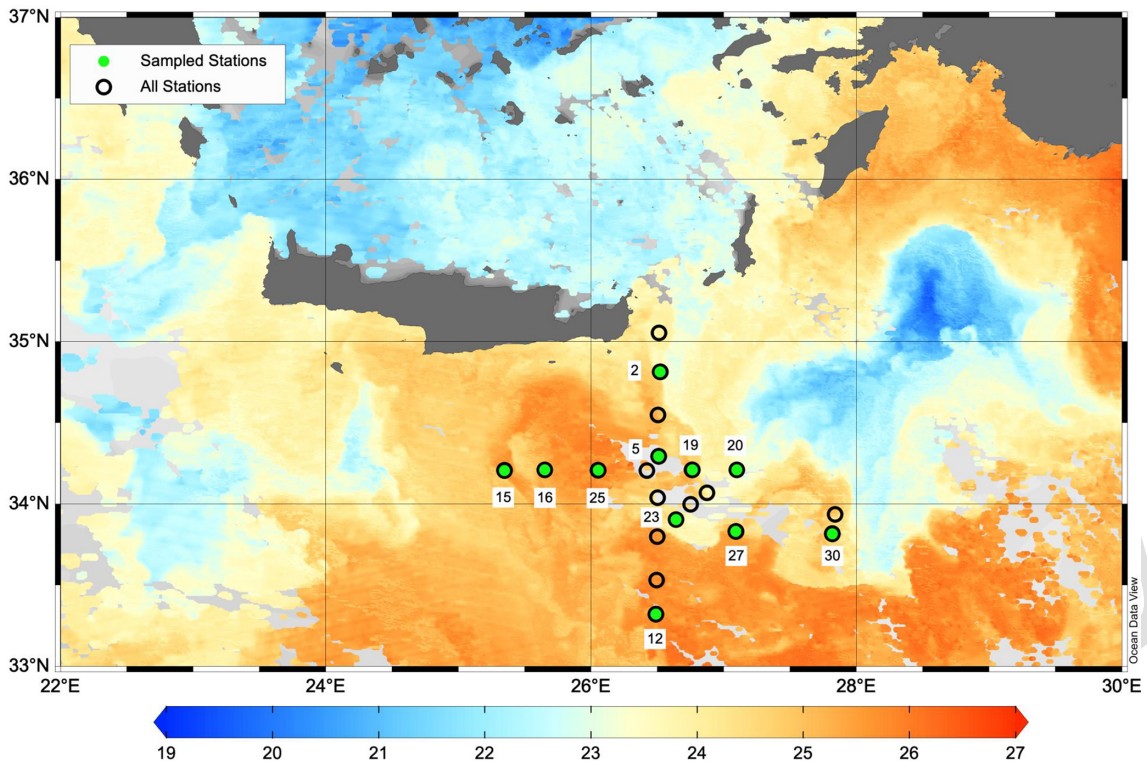

**Figure 1.** Map of the PERLE1 cruise southeast of Crete in October 2018. Sampled stations are indicated with green dots. Background image is the sea surface temperature (L3S ultra-high-resolution product distributed by the Copernicus Marine Environment Monitoring Service; Buongiorno Nardelli et al., 2013) on 16 October 2018. The warm core Ierapetra anticyclone is observed around $34°30'$ N, $26°$ E. Map was created using Ocean Data View v5.6 (Schlitzer, 2018).

sive investigation of the western Levantine Sea carried out in the framework of the French program MISTRALS MER-MeX (Marine Mediterranean eXperiment) and its component PERLE (Pelagic Ecosystem Response to dense water for-
5 mation in the Levant Experiment; D'Ortenzio et al., 2021). During the autumn cruise, PERLE1, the sampling plan focused on the warm core of the anticyclone Ierapetra and its extensions in the Levantine Basin (Fig. 1). During the winter cruise, PERLE2, the sampling plan extended over the whole
10 area and a larger panel of dynamical features were encountered including the cold core of the Rhodes cyclonic gyre located east of Crete (Fig. 2; more details in Taillandier et al., 2022).

For the purpose of this study, we sampled 11 stations dur-
15 ing PERLE1 and 14 stations during PERLE2, corresponding to a large variety of hydrological situations (Table 1). For both cruises, full-depth oceanographic stations were carried out using a CTD (conductivity–temperature–depth) rosette equipped with a sampling system of 24 Niskin bottles and a
20 Sea-Bird SBE 9plus underwater unit equipped with pressure, temperature, conductivity, oxygen and chlorophyll fluorescence sensors.

Water subsamples from the Niskin bottles were taken for nutrients (inorganic and organic, including nanomolar anal-

yses of DIP and $L_{DOP}$), biological stocks (flow cytometry 25 counts and chlorophyll $a$) and phosphatase activities (PME and PDE).

At each station some selected layers were sampled: 10 layers for DIP with the sensitive technique and $L_{DOP}$ (between 0 and 200 m during PERLE1 and 0 and 300 m during PERLE2, 30 in connection with nutrient and chlorophyll distributions in epipelagic water column) and, among these, 6 layers for PME and PDE activities. Other nutrient analyses (nitrate, nitrite, DOP, DIP with the classical method) were sampled between the surface and the bottom depth: 12 levels between 35 the surface and 300 m depth and 6 levels below 300 m depth. However only the 0–300 m layer is described in this study.

## 2.2 Nutrients

Seawater samples for standard nutrient analysis were filtered online (0.45 µm cellulose acetate filters) directly from 40 the Niskin bottles in 20 mL acid-washed polyethylene vials; these were stored frozen until analysis for PERLE1 and immediately analyzed on board for PERLE2. Micromolar nutrient concentrations of nitrate, nitrite and phosphate were determined by colorimetry (Aminot and Kérouel, 2007) us- 45 ing a segmented flow analyzer (SEAL Analytical AutoAna-

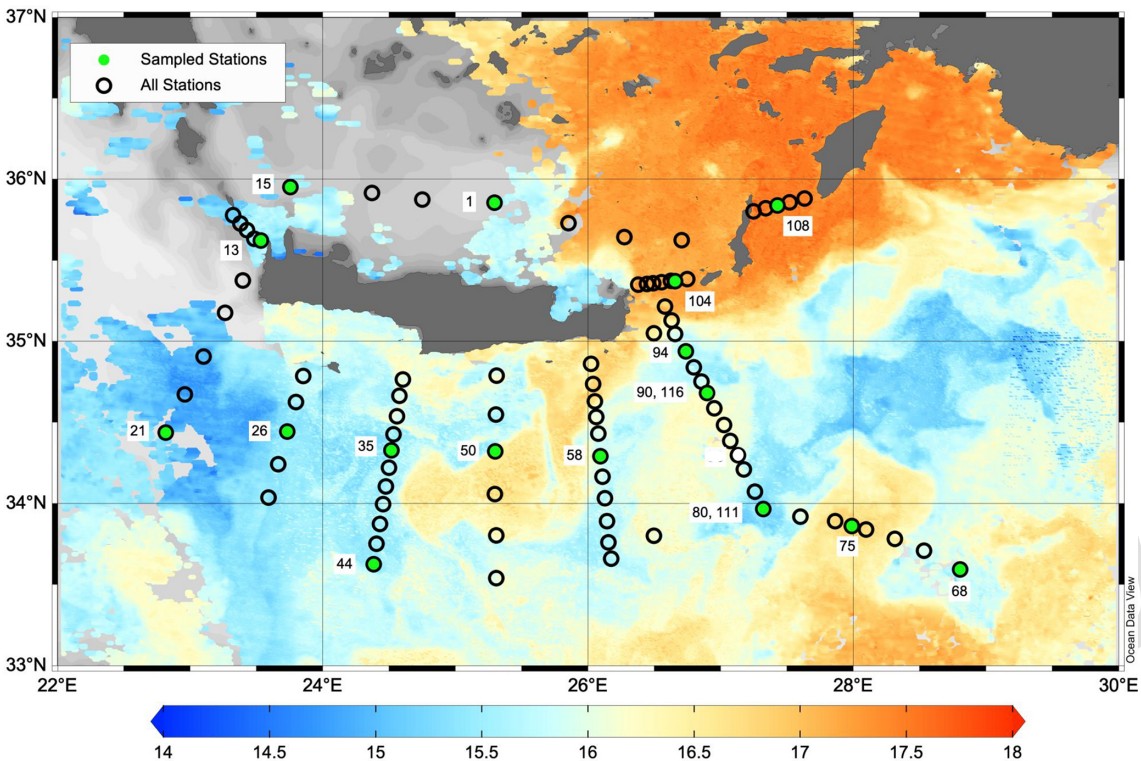

**Figure 2.** Map of the PERLE2 cruise surrounding Crete in February–March 2019. Sampled stations are indicated with green dots. Background image is the sea surface temperature (L3S ultra-high-resolution product distributed by the Copernicus Marine Environment Monitoring Service; Buongiorno Nardelli et al., 2013) on 4 March 2019. The cold core of the Rhodes cyclonic gyre is observed around 35° N, 29° E. Map was created using Ocean Data View v5.6 (Schlitzer, 2018).

lyzer 3 HR; formerly Bran+Luebbe), with analytical precision of 0.02, 0.01 and 0.01 µM, respectively.

Samples for the determination of nanomolar concentrations of DIP were collected in HDPE (high-density polyethylene) bottles previously cleaned with superpure HCl after filtration through 0.2 µm. During PERLE1, samples were stored frozen until analysis in the laboratory. During PERLE2, samples were analyzed on board immediately after sampling. Nanomolar DIP was analyzed using the LWCC (liquid waveguide capillary cell) method modified from Zhang and Chi (2002), with a detection limit of 1 nM.

Total dissolved phosphorus (TDP) was measured using the segmented flow analyzer technique after high-temperature (120 °C) persulfate wet oxidation mineralization (Pujo-Pay et al., 1997; Pujo-Pay and Raimbault, 1994). Dissolved organic phosphorus (DOP) was obtained as the difference between TDP and DIP.

The monoesterase hydrolysable fraction of DOP ($L_{DOP}$) was estimated after enzymatic hydrolysis of the $< 0.2$ µm filtrate in the presence of a purified phosphatase alkaline (AP) enzyme from *Escherichia coli* (Sigma P4252) (Djaoudi et al., 2018a) in HDPE bottles. The AP was diluted with pure water to prepare a working solution of 0.2 units mL$^{-1}$. Equal volumes (0.6 mL) of AP working solution and Tris buffer

(0.5 M, pH 8) were added to 30 mL of the $< 0.2$ µm filtered samples. Samples were then incubated for 3 h in the dark at 30 °C. The duration of the incubation and the hydrolysis efficiency was checked with glucose 6-phosphate. After incubation, samples were stored frozen until analysis (PERLE1) or analyzed on board (PERLE2). $L_{DOP}$ was obtained as the difference in DIP concentration before and after incubation. A blank was run at each station consisting of 30 mL ultrapure water in which 0.6 mL of working AP solution and Tris buffer was introduced.

## 2.3 Biological stocks and fluxes

Flow cytometry was used for the enumeration of autotrophic prokaryotic and eukaryotic cells, heterotrophic prokaryotes (Hprok), and heterotrophic nanoflagellates (HNF). Water samples (4.5 and 2 mL) were fixed with 25 % glutaraldehyde grade I (1 % final concentration), flash-frozen and stored at $-80$ °C until analysis. For Hprok, the 2 mL samples were defrosted at room temperature and subsequently analyzed using a FACSCanto flow cytometer (BD Biosciences) of the BioPIC (Biology Platform of Imaging and flow Cytometry) platform (https://www.obs-banyuls.fr/fr/rechercher/plateformes/ biopic.html, last access: 5 May 2024), equipped with fiber-

**Table 1.** Position, sampling date, and some physical and biogeochemical characteristics of the stations studied during the PERLE1 (October 2018) and PERLE2 (February–March 2019) cruises. Lat: latitude, Long: longitude, SST: sea surface temperature, MLD: mixed-layer depth, Z Pcline: phosphacline depth, Z Ncline: nitracline depth, Tchla: total chlorophyll $a$.

| Cruise | Station | Date and time (UTC) | Lat (° N) | Long (° E) | SST (°C) | MLD (m) | Z Pcline (m) | Z Ncline (m) | Integrated Tchla (mg m$^{-2}$) |
|---|---|---|---|---|---|---|---|---|---|
| PERLE1 | 2 | 10 Oct 2018 20:39 | 34.82 | 26.52 | 24.9 | 38 | 106 | 126 | 16.1 |
| PERLE1 | 5 | 11 Oct 2018 10:36 | 34.29 | 26.51 | 25.1 | 48 | 188 | 154 | 14.6 |
| PERLE1 | 12 | 12 Oct 2018 08:43 | 33.32 | 26.49 | 26.2 | 22 | 119 | 105 | 13.4 |
| PERLE1 | 15 | 15 Oct 2018 03:11 | 34.21 | 25.95 | 25.3 | 30 | 145 | 123 | 13.1 |
| PERLE1 | 16 | 15 Oct 2018 08:38 | 34.21 | 25.65 | 25.5 | 25 | 193 | 143 | 11 |
| PERLE1 | 19 | 16 Oct 2018 00:00 | 34.21 | 26.76 | 24.9 | 23 | 175 | 137 | 21.4 |
| PERLE1 | 20 | 16 Oct 2018 04:40 | 34.21 | 27.10 | 23.9 | 36 | 122 | 90 | 15 |
| PERLE1 | 23 | 16 Oct 2018 17:09 | 33.90 | 26.64 | 24.9 | 35 | 167 | 123 | 23.8 |
| PERLE1 | 25 | 18 Oct 2018 01:29 | 34.21 | 26.05 | 25.9 | 88 | 239 | 185 | 20.9 |
| PERLE1 | 27 | 19 Oct 2018 23:08 | 33.83 | 27.09 | 24.1 | 19 | 100 | 83 | 16.5 |
| PERLE1 | 30 | 20 Oct 2018 08:10 | 33.82 | 27.82 | 24.7 | 18 | 135 | 98 | 16.8 |
| PERLE2 | 1 | 27 Feb 2019 08:33 | 35.86 | 25.30 | 16.0 | 145 | 82 | 60 | 51.7 |
| PERLE2 | 13 | 28 Feb 2019 17:01 | 35.62 | 23.54 | 15.8 | 118 | 247 | 188 | 59.2 |
| PERLE2 | 15 | 1 Mar 2019 01:44 | 35.95 | 23.76 | 15.6 | 269 | 164 | 90 | 66.2 |
| PERLE2 | 21 | 2 Mar 2019 03:22 | 34.44 | 22.82 | 15.5 | 27 | 115 | 17 | 33.2 |
| PERLE2 | 26 | 2 Mar 2019 22:30 | 34.44 | 23.73 | 15.6 | 113 | 94 | 70 | 39.8 |
| PERLE2 | 35 | 4 Mar 2019 02:27 | 34.33 | 24.52 | 15.6 | 53 | 164 | 144 | 52.6 |
| PERLE2 | 44 | 5 Mar 2019 03:49 | 33.62 | 24.38 | 15.7 | 61 | 67 | 24 | 28.2 |
| PERLE2 | 50 | 6 Mar 2019 02:10 | 34.32 | 25.30 | 16.6 | 213 | 258 | 211 | 69.3 |
| PERLE2 | 58 | 7 Mar 2019 05:23 | 34.29 | 26.09 | 16.0 | 97 | 151 | 101 | 60.9 |
| PERLE2 | 68 | 8 Mar 2019 20:50 | 33.59 | 28.81 | 16.4 | 34 | 89 | 40 | 40.3 |
| PERLE2 | 75 | 10 Mar 2019 01:10 | 33.86 | 27.99 | 16.7 | 45 | 73 | 11 | 48.1 |
| PERLE2 | 80 | 10 Mar 2019 15:40 | 33.96 | 27.32 | 16.4 | 14 | 37 | 2 | 65.6 |
| PERLE2 | 90 | 12 Mar 2019 01:03 | 34.68 | 26.90 | 16.6 | 25 | 82 | 64 | 45.6 |
| PERLE2 | 94 | 12 Mar 2019 13:03 | 34.94 | 26.74 | 16.8 | 19 | 15 | 0 | 26.5 |
| PERLE2 | 104 | 13 Mar 2019 06:32 | 35.37 | 26.66 | 17.4 | 63 | 177 | 134 | 39.5 |
| PERLE2 | 108 | 13 Mar 2019 17:14 | 35.84 | 27.43 | 17.4 | 103 | 265 | 240 | 67.4 |
| PERLE2 | 111 | 14 Mar 2019 13:29 | 33.96 | 27.32 | 16.1 | 53 | 45 | 32 | 63.8 |
| PERLE2 | 116 | 15 Mar 2019 02:56 | 34.68 | 26.90 | 16.2 | 48 | 104 | 5 | 38.9 |

optic-emitted light (405, 488 and 633 nm). Fluorescent 1 µm beads for Hprok and 10 µm beads for HNF (Polysciences, Inc., Europe) were added to each sample as an internal standard to normalize cell properties and to compare cell populations. Accurate analyzed volumes and subsequent estimations of cell concentrations were calculated using BD Trucount™ beads. Cells of Hprok and HNF were discriminated and enumerated according to their right-angle light-scattering properties (side scatter, SSC; roughly related to cell internal complexity) and green (515–545 nm) fluorescence due to nucleic acid staining with SYBR Green I (Molecular Probes) for 15 min at room temperature in the dark (Marie et al., 1997). Hprok were enumerated as the sum of two clusters (bacteria with high nucleic acid (HNA) and low nucleic acid (LNA) content). Biomass of Hprok (Hprok-C) was calculated assuming 10 fg C per cell. The whole population of HNF was also discriminated following the same principle (Christaki et al., 2011).

Phytoplankton samples were analyzed according to Marie et al. (2000) protocols using FACSCalibur (BD Biosciences®) of the PRECYM flow cytometry platform (https://precym.mio.osupytheas.fr/, last access: 5 May 2024), equipped with a blue (488 nm) laser and a red (634 nm) laser. Just before phytoplankton analyses, 2 µm beads were added as an internal standard and to discriminate picoplankton ($< 2$–3 µm) and nanoplankton ($> 2$–3 µm) populations (Fluoresbrite YG, Polysciences, Inc.). A solution of Trucount beads (BD Biosciences®) was also added to the samples to determine the volume analyzed. The same sample was acquired twice using two different settings: the first one to assess picophytoeukaryotes (Picoeuk), nanophytoeukaryotes (Nanoeuk) and cryptophyte-like cells (Crypto) and the second one, using a higher amplification of the photodetector of the red fluorescence signal (induced by chlorophyll), to focus on the small size and/or cells with low chlorophyll $a$ fluorescence, such as *Prochlorococcus* (Pro) and *Synechococcus*

(Syn). The cell concentration was determined from both Trucount beads and flow rate measurements.

Total chlorophyll *a* (Tchla) is the sum of chlorophyll *a* and divinyl chlorophyll *a*. It was calculated by high-performance liquid chromatography (HPLC) analysis after the extraction of pigments from GF/F filters (Ras et al., 2008). The fluorescence sensor was calibrated with Tchla. The total phytoplankton biomass (phyto C) was calculated assuming an overall C : Tchla ratio of 50.

Ectoenzymatic activities were measured fluorometrically with fluorogenic model substrates (Hoppe, 1983), using 4-methylumbelliferyl phosphate (MUF-P, Sigma) and bis(4-methylumbelliferyl)phosphate (bis-MUF-P, Chem-Impex) to assess phosphomonoesterase (PME) and phosphodiesterase (PDE) activities, respectively. The release of MUF from fluorogenic substrates was monitored by measuring the increase in fluorescence in the dark (excitation/emission of 365/450 nm, wavelength bandwidth of 5 nm) periodically (at least five times) for up to 12 h using a Varioskan LUX microplate reader. The 24-well microplates were incubated in a thermostatic incubator at the in situ temperature in the dark. Aliquots (2 mL) of samples were incubated with final concentrations of fluorogenic substrates varying from 0.025 to 1 μM for MUF-P and from 0.025 to 50 μM for bis-MUF-P. These ranges were chosen after preliminary tests using 0.025–50 μM concentrations for both activities. The parameters Vm (maximum hydrolysis velocity) and Km (Michaelis–Menten constant that reflects enzyme affinity for the substrate) as well as their corresponding errors were estimated by non-linear regression (using the PRISM software, https://www.graphpad.com/features, last access: 5 May 2024) using the Michaelis–Menten equation.

$$V = \frac{\text{Vm} \times S}{\text{Km} + S} \tag{1}$$

Here $V$ is the hydrolysis rate and $S$ is the fluorogenic substrate concentration added. Turnover times (TTs) were calculated as the Km : Vm ratio.

## 2.4 Data processing and diagnostics

Measurements by CTD sensors were processed into 1 m resolution vertical profiles for in situ temperature, salinity, potential density anomaly referenced to the surface (shortened to density hereinafter) and calibrated chlorophyll fluorescence. The mixed-layer depth (MLD) was determined as in Taillandier et al. (2022). The depth of the nutriclines were calculated from DIP and $NO_x$ (nitrate + nitrate) vertical profiles. The nitracline depth (Ncline) was estimated by the intercept of the regression line reported in a diagram of $NO_x$ vs. depth, and the phosphacline (Pcline) was shown by the intercept of the regression line reported in a diagram of DIP vs. depth, in which we used the DIP concentrations determined with the LWCC technique for depleted layers and classical DIP measurements for richer layers (> 0.08 μM). The least-squares regressions were made on the linear parts of the plots of $NO_x$ and DIP vs. depth.

## 3 Results

### 3.1 Distribution of physical properties

During the wintertime case (PERLE2 cruise), sea surface temperature ranged from 15.5 to 17.4 °C (Table 1). The vertical gradient of temperature between the surface and 300 m depth was weak (maximum difference at a temperature of 2.4 °C). Density profiles (Fig. S1 in the Supplement) showed (i) very well-mixed surface layers at station (st) 1, 13 and 15 – sampled at the beginning of the cruise in the Cretan Sea including the Kythira Strait – and at st 50, located south of Crete in the center of an anticyclonic gyre (Fig. S1b); (ii) slightly mixed conditions at st 104–108 located in the Kasos and Karpathos straits along the anticyclonic side of the geostrophic jets entering in the Cretan Sea; and (iii) variable degrees of stratification during the progression of the cruise toward the east and the Rhode cyclonic gyre at the other stations of the PERLE2 cruise (Fig. S1c).

The MLD varied over a large range, between 14 and 269 m (mean ± SD of 83 ± 69, Table 1). The highest MLDs were encountered in the Cretan Sea (st 1, 13, 15) and in the center of an anticyclone south of Crete (st 50, Figs. 2 and S1b). In contrast, some stations sampled along the easternmost transect (e.g., st 80 and 94) were located in extensions of the Rhodes cyclonic gyre and presented lower MLDs (Figs. 2 and S1c, Table 1).

During the autumn case (PERLE1 cruise), density profiles were more similar among stations because the stations were sampled in a more restricted area within the anticyclone Ierapetra (Fig. 1). Sea surface temperatures ranged from 26.5 to 27.8 °C (Table 1), with an important thermal and density stratification (Fig. S1a). The mean MLD (35 ± 19 m, Table 1) was significantly lower than in the winter case (Mann–Whitney test, $p = 0.012$).

### 3.2 Nutrients

In winter, the depth of the Pcline was on average 124 ± 76 m, showing a great variability among stations (Fig. 3b and c). In autumn, vertical distributions of DIP showed depleted values in the mixed layer and a rapid increase with depth, with Pcline depths being more homogeneous than in winter, and reaching on average 154 ± 43 m (Table 1, Fig. 3a), although the difference between both cruises was not statistically different ($p = 0.10$). The same trend was observed for vertical profiles of $NO_x$ (the sum of nitrate + nitrite): greater variability in winter and homogeneous profiles of concentrations in autumn (Fig. 3d–f), with however significant deepening in autumn ($p = 0.044$) and Ncline depth being reached at 124 ± 30 m in autumn vs. 80 ± 75 m in winter. In winter, the deepest Ncline and Pcline (Table 1) were observed at the

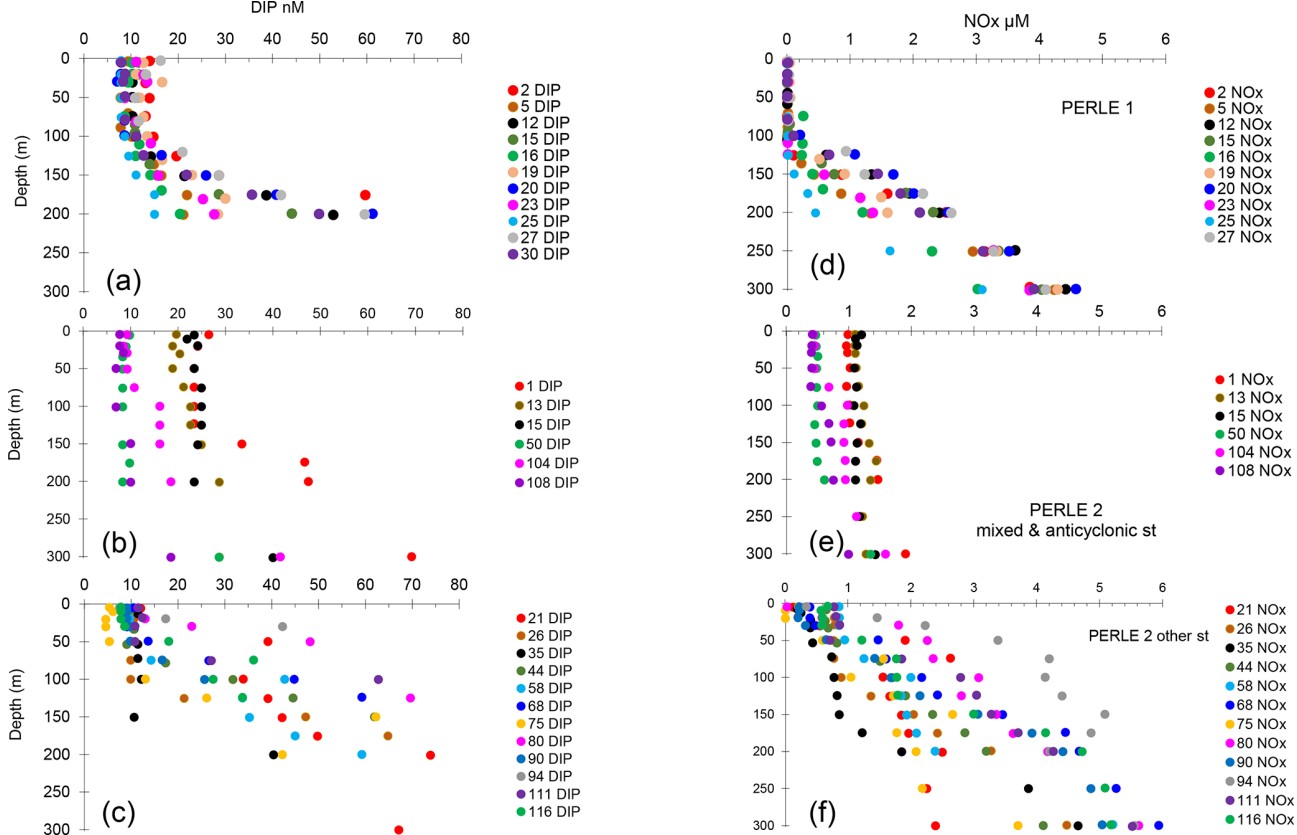

**Figure 3. (a–c)** Vertical distributions of dissolved inorganic phosphorus (DIP). **(d–f)** Vertical distributions of the sum of nitrate + nitrite concentrations ($NO_x$). **(a, d)** PERLE1 cruise (October 2018); **(b, e)** mixed and anticyclonic stations of the PERLE2 cruise (February–March 2019); **(c, f)** other stations of the PERLE2 cruise.

stations situated within anticyclonic areas (st 50, 108) followed by the well-mixed conditions at the beginning of the cruise (st 1, 13, 15). The shallowest Pcline and Ncline were reached at st 80 and 94. In all cases (except one), Pcline was deeper than Ncline, with an average difference of $44 \pm 26$ m in winter and $29 \pm 21$ m in autumn (statistically not different; Mann–Whitney test, $p > 0.05$).

During the winter cruise, DIP in the ML was higher in the Cretan Sea (average concentration of 20–24 nM at st 1, 13 and 15, Table 2). DIP in the ML was on the same order of magnitude in the rest of the stations of the winter cruise (8–12 nM) and at all the stations during the autumn cruise (8–13 nM). Consequently, DIP concentrations were not particularly lower within the ML in the autumn oligotrophic case (Fig. 3a), and DIP concentrations between PERLE1 and PERLE2 were not statistically different ($p = 0.6$). This was not the case for $NO_x$, which clearly showed N-depleted conditions within the ML in autumn (with means significantly lower than in winter, $p < 0.001$), with frequent subsurface data below the threshold of detection of 0.01 μM (mean concentrations in the ML ranged 0.01–0.03 μM, Table 2, Fig. 3d). In winter, $NO_x$ varied within a large range, with means of concentrations per station inside the ML be-

ing higher in the Cretan Sea (st 1, 13, 15: 0.99–1.13 μM) and anticyclonic areas (st 111, 116: 0.79–0.61 μM). On the other hand, $NO_x$ showed lower values in the ML at some other stations mostly situated on the easternmost transect (st 35, 75, 90, 94: 0.22–0.32 μM) (Table 2, Fig. 3f). The ratio of $NO_x$ to DIP within the ML was significantly ($p < 0.001$) higher in the winter case ($55 \pm 19$ vs. $1.8 \pm 0.7$, Fig. S2).

For the whole data set, the ranges of DOP and $L_{DOP}$ were 8–92 and 1–17 nM, respectively, during autumn and 10–120 and 2–64 nM, respectively, during winter (Fig. 4a–c). The higher values of $L_{DOP}$ ($> 25$ nM) were encountered in a few cases during winter in the Cretan Sea, and at st 75, for the remaining data set, values were all below 25 nM (Fig. S3). For the whole data set, DIP and DOP explained the lack of or low variability in $L_{DOP}$ (Fig. S3, $r^2 < 0.06$ for all tests on log-transformed data). For the whole data set, the fraction of $L_{DOP}$ in DOP (%$L_{DOP}$) varied within a large range, from 1.3 % to 97 %, with a mean of $28\% \pm 18\%$.

Within the ML, means of DOP and $L_{DOP}$ per station were significantly lower during the autumn cruise: DOP means being $25 \pm 10$ vs. $50 \pm 16$ nM ($p < 0.001$) and $L_{DOP}$ means being $6 \pm 2$ vs. $16 \pm 9$ nM ($p < 0.001$).

**Table 2.** Mean abundances of the main flow cytometric groups and nutrient concentrations inside the ML at each station. Hprok: heterotrophic prokaryotes, Proc: *Prochlorococcus*, Syn: *Synechococcus*, Picoeuk: picophytoeukaryotes, Nanoeuk: nanophytoeukaryotes, DIP: dissolved inorganic phosphorus, $NO_x$: sum of nitrate + nitrite, DOP: dissolved organic phosphorus, $L_{DOP}$: labile dissolved organic phosphorus, $NO_x$ : DIP: ratio of $NO_x$ to DIP, NA: not available.

| Cruise | Station | Hprok ($10^5$ mL$^{-1}$) | Proc ($10^3$ mL$^{-1}$) | Syn ($10^3$ mL$^{-1}$) | Picoeuk ($10^3$ mL$^{-1}$) | Nanoeuk (mL$^{-1}$) | DIP (nM) | $NO_x$ (µM) | DOP (nM) | $L_{DOP}$ (nM) | $L_{DOP}$ (%) | $NO_x$ : DIP (ratio) |
|---|---|---|---|---|---|---|---|---|---|---|---|---|
| PERLE1 | 2 | 3 | 0.87 | 13.1 | 0.42 | 90 | 13 | 0.011 | 21 | 4 | 19 | 0.8 |
| PERLE1 | 5 | 3.2 | 0.57 | 11.3 | 0.4 | 117 | 9 | 0.012 | 19 | 6 | 34 | 1.3 |
| PERLE1 | 12 | 3.1 | 0.11 | 6 | 0.29 | 64 | 10 | 0.01 | 16 | 7 | 46 | 0.9 |
| PERLE1 | 15 | 3.8 | 0.59 | 13.6 | 0.62 | 125 | 12 | 0.026 | 18 | 2 | 11 | 2.3 |
| PERLE1 | 16 | 3.9 | 0.48 | 10.3 | 0.37 | 56 | 10 | 0.01 | 40 | NA | NA | 1 |
| PERLE1 | 19 | 3.2 | 0.62 | 11.1 | 0.42 | 54 | 12 | 0.035 | 21 | 3 | 16 | 2.9 |
| PERLE1 | 20 | 3.8 | 3.1 | 14 | 0.62 | 73 | 8 | 0.01 | 36 | 6 | 17 | 1.3 |
| PERLE1 | 23 | 3.6 | 0.68 | 11.9 | 0.58 | 105 | 12 | 0.012 | 47 | 5 | 10 | 1 |
| PERLE1 | 25 | 3.1 | 0.49 | 13.1 | 0.55 | 83 | 8 | 0.011 | 18 | 8 | 46 | 1.4 |
| PERLE1 | 27 | 3.4 | 2.6 | 11.5 | 0.48 | 84 | 16 | 0.024 | 22 | 6 | 28 | 1.5 |
| PERLE1 | 30 | 2.7 | 0.32 | 8.3 | 0.39 | 52 | 8 | 0.019 | 22 | 10 | 46 | 2.5 |
| PERLE2 | 1 | 3 | 0.38 | 0.35 | 0.09 | 13 | 24 | 0.99 | 69 | 31 | 44 | 41 |
| PERLE2 | 13 | 3.6 | 0.87 | 0.86 | 0.07 | 15 | 20 | 1.13 | 62 | 37 | 59 | 56 |
| PERLE2 | 15 | 3.9 | 0.9 | 0.71 | 0.24 | 13 | 24 | 1.13 | 74 | 11 | 15 | 48 |
| PERLE2 | 50 | 4.2 | 1.2 | 1.8 | 0.06 | 14 | 8 | 0.5 | 71 | 9 | 14 | 58 |
| PERLE2 | 104 | 3.9 | 1.2 | 2.9 | 0.17 | 28 | 9 | 0.45 | 26 | 16 | 61 | 50 |
| PERLE2 | 108 | 4.1 | 0.68 | 1.9 | 0.09 | 25 | 7 | 0.44 | 22 | 13 | 44 | 59 |
| PERLE2 | 21 | 3.4 | 1.8 | 2.3 | 0.06 | 20 | 12 | 0.17 | 44 | 5 | 12 | 14 |
| PERLE2 | 26 | 4.4 | 2.1 | 2.2 | 0.06 | 21 | 10 | 0.8 | 49 | 9 | 18 | 78 |
| PERLE2 | 35 | 3.3 | 1.7 | 2.4 | 0.08 | 25 | 10 | 0.32 | 62 | 7 | 13 | 31 |
| PERLE2 | 44 | 3.2 | 2 | 1.9 | 0.07 | 14 | 9 | 0.7 | 56 | 14 | 25 | 74 |
| PERLE2 | 58 | NA | 1.7 | 1.9 | 0.24 | 43 | 11 | 0.96 | 39 | 12 | 31 | 87 |
| PERLE2 | 68 | 3.6 | 4.3 | 3.1 | 0.22 | 46 | 9 | 0.43 | 34 | 11 | 31 | 46 |
| PERLE2 | 75 | 2.9 | 1.1 | 2.8 | 0.06 | 27 | 5 | 0.22 | 56 | 29 | 65 | 41 |
| PERLE2 | 80 | 6.1 | 1.7 | 3.8 | 0.27 | 48 | 11 | 0.034 | 47 | 12 | 26 | 3 |
| PERLE2 | 90 | 4.5 | 3.4 | 5.4 | 0.17 | 49 | 9 | 0.22 | 55 | 16 | 29 | 25 |
| PERLE2 | 94 | 6.1 | 3.1 | 5.9 | 0.3 | 31 | 7 | 0.34 | 57 | 19 | 33 | 45 |
| PERLE2 | 111 | 7.1 | 5.1 | 4.6 | 0.26 | 22 | 11 | 0.79 | 42 | 15 | 38 | 70 |
| PERLE2 | 116 | 6.6 | 2.9 | 3.6 | 0.19 | 23 | 8 | 0.61 | 25 | 13 | 54 | 76 |

## 3.3 Chlorophyll stocks and phytoplankton populations

In autumn, the vertical distribution of chlorophyll stocks was homogeneous, showing low values for the surface (0.02–0.07 µg Tchla L$^{-1}$) and deep chlorophyll maximum (DCM) visible around 84–125 m depth, which peaked up to 0.2 µg Tchla L$^{-1}$ (Fig. S1d). In autumn, integrated chlorophyll stocks were low ($16 \pm 4$ mg Tchla m$^{-2}$). In winter, at stations under deep ML conditions, Tchla stayed homogeneous down to 300 m (st 1, 13 15 and 50) and showed a small decrease with depth at st 104 and 108 (Fig. S1e). In other stations of the winter cruise, diverse shapes of the Tchla vertical distribution were seen, with surface or subsurface peaks varying from 0.25 (st 21) up to 0.95 mg Tchla L$^{-1}$ (st 80). Integrated stocks were on average significantly higher during the winter cruise ($50 \pm 14$ mg Tchla m$^{-2}$, $p < 0.001$), showing a greater variability than in the autumn cruise, with maximum values ($> 60$ mg Tchla m$^{-2}$) reached at st 15 and 50 (mixed stations) and at some other stations sampled at the end of the cruise and/or under influence of the Rhodes cyclonic gyre or anticyclonic influence (st 58, 80, 108, 111, Fig. S1f, Table 1).

In autumn, all picophytoplankton groups were more abundant than in winter. *Prochlorococcus* abundances peaked within the DCM depth with maxima varying according to the station between 23 and $47 \times 10^3$ cells mL$^{-1}$ (Fig. S4a), whereas picophytoeukaryotes were rather peaking at the surface ($0.13–0.68 \times 10^3$ cells mL$^{-1}$, Fig. S5a) and *Synechococcus*-like abundances were peaking within the subsurface layers ($7.9–19 \times 10^3$ cells mL$^{-1}$, Fig. S4d). Heterotrophic prokaryotes also peaked within the DCM depth with abundances at the peak ranging from $3.6–5.4 \times 10^5$ cells mL$^{-1}$ (Fig. S6d). In winter, following mixing/stratification conditions, all phytoplankton groups (Syn, Proc, Picoeuk, Nanoeuk, Crypto) as well as Hprok were low and relatively homogeneous along the vertical profile at st 1, 13, 15, 50, 104 and 108 (Figs. S4b and e, S5b and e, and S6b and e). Proc showed variable profiles for the other stations of the PERLE2 cruise, with surface or subsurface peaks (st 68, 90, 111, Fig. S4c). Syn followed Proc vertical trends also and peaked at the surface or within the subsurface (Fig. S4f). Ver-

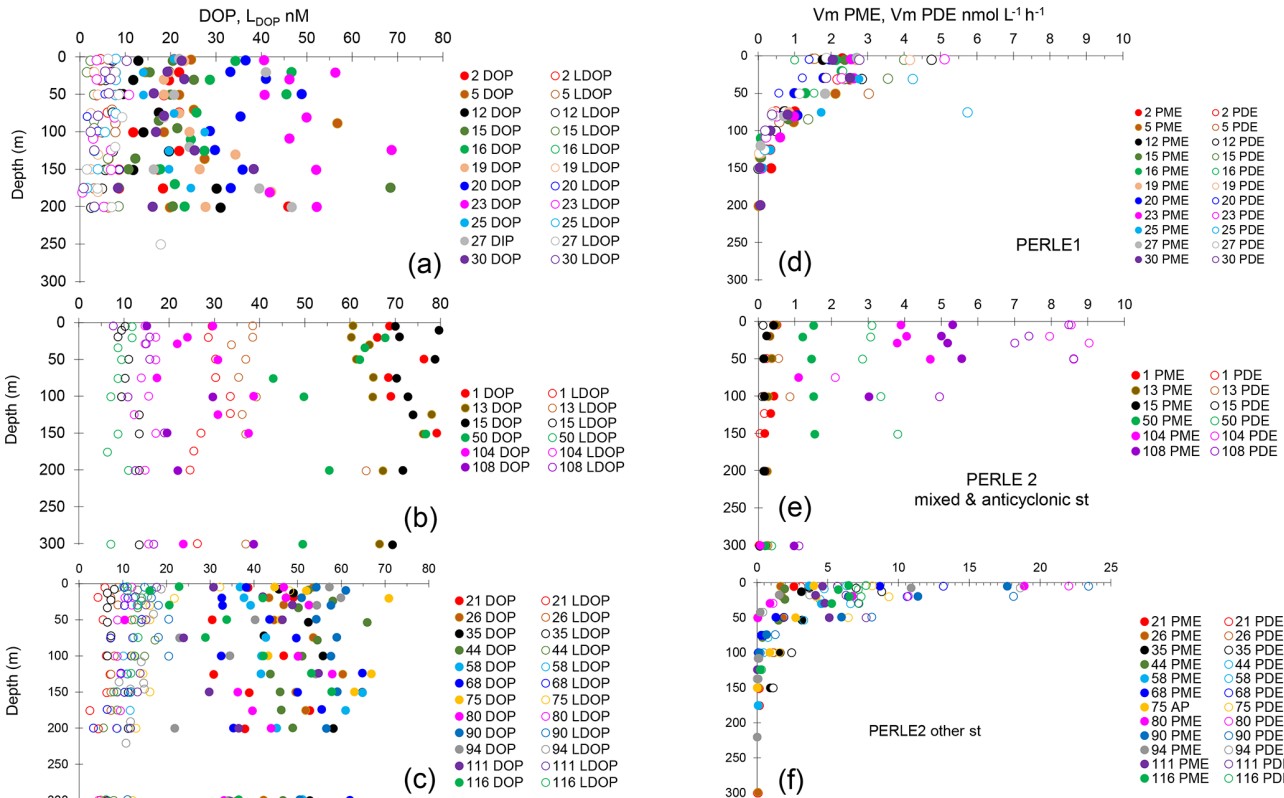

**Figure 4. (a–c)** Vertical distributions of dissolved organic phosphorus (DOP) and labile dissolved organic phosphorus (L$_{DOP}$). **(d–f)** Vertical distributions (0–300 m) of maximum hydrolysis rates of phosphomonoesterase (Vm PME) and phosphodiesterase (Vm PDE). **(a, d)** PERLE1 cruise (October 2018); **(b, e)** mixed and anticyclonic stations of the PERLE2 cruise (February–March 2019); **(c, f)** other stations of the PERLE2 cruise.

tical distribution of abundance of Picoeuk also varied along the different profiles, peaking between the surface and 100 m (Fig. S5c). Nanoeuk followed the same vertical trends as Picoeuk, with maximum abundances up to 140 cells mL$^{-1}$ reached during the PERLE1 cruise (Fig. S5f). Cryptophyte-like cells were scarce but notably showed small surface abundance peaks during the winter cruise at some specific stations (st 80, 90, Fig. S6c). Finally, vertical profiles of Hprok (Fig. S6d–f) also varied in shape and the order of magnitude of abundances reached, with abundances of Hprok at the peak ranging from $3.2$–$7.3 \times 10^5$ cells mL$^{-1}$.

### 3.4 Phosphomonoesterase and phosphodiesterase activities

For PME, 20 kinetics and, for PDE, 41 kinetics of 174 were not available due to the low increase in fluorescence with time after the addition of low concentrations of MUF-P or bis-MUF-P. These were generally situated within the deepest layers sampled. After testing a large set of substrate concentrations between 25 nM and 50 µM on some samples, the saturation state was reached at different concentrations for PME and PDE (Fig. 5). PME reached its maximum activities (Vm)

after the addition of 1 µM MUF-P, whereas it was necessary to add up to 50 µM bis-MUF-P to reach the saturation state with PDE. Consequently, the affinity constants (Km) were higher for PDE (Fig. 6a). On average for the whole data set, the Km PDE was 33-fold higher than Km PME (mean $\pm$ SD: $33 \pm 25$); however Km PME and Km PDE were not correlated. Km PDE decreased with depth during both cruises (Fig. 6d and f), except at the well-mixed stations 1, 13 and 15 in February–March (Fig. 6e). On the other hand, Km PME either increased with depth (autumn cruise, Fig. 6a) or did not vary with depth (winter cruise, Fig. 6b and c). Neither Km PME nor Km PDE correlated with DOP or L$_{DOP}$, whatever the cruise (log–log relationships tested, $p > 0.05$). On the other hand, Km showed variable correlations with DIP depending on the cruise or the enzyme: Km PDE decreased when DIP increased during both cruises, and Km PME increased when DIP increased in October, whereas this relation was insignificant in February–March (Fig. 7a).

Within the ML, the Km mean per station ranged from 1.10 to 7.58 µM for PDE and from 0.054 to 0.288 µM for PME (Table 3). Km PME within the ML was significantly different in the PERLE1 and PERLE2 cruises ($0.066 \pm 0.008$ µM and $0.169 \pm 0.060$ nM; Mann–Whitney

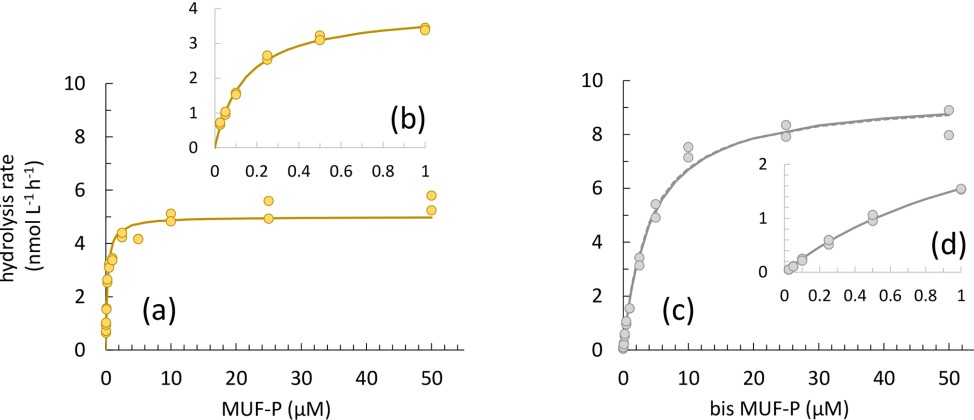

**Figure 5. (a)** Example of a concentration kinetic with an MUF-P addition, **(b)** with a focus on the 0–1 μM range. **(c)** Example of a concentration kinetic with a bis-MUF-P addition, **(d)** with a focus on the 0–1 μM range.

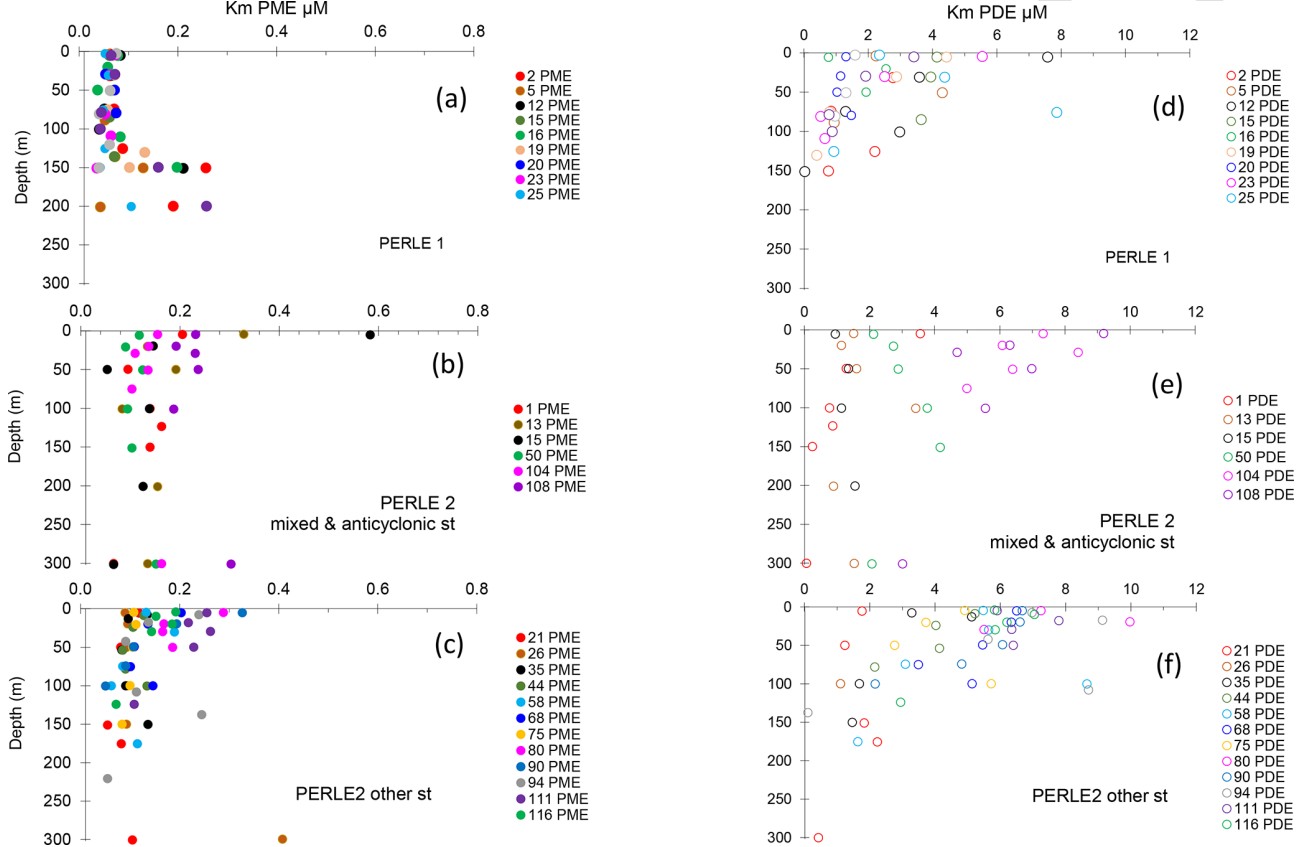

**Figure 6. (a–c)** Vertical distributions of Km PME. **(d–f)** Vertical distributions of Km PDE. **(a, d)** PERLE1 cruise (October 2018); **(b, e)** mixed and anticyclonic stations of the PERLE2 cruise (February–March 2019); **(c, f)** other stations of the PERLE2 cruise.

test, $p < 0.001$). This difference was insignificant ($p = 0.06$) for Km PDE ($3.45 \pm 1.85\,\mu\mathrm{M}$ and $4.56 \pm 2.25\,\mu\mathrm{M}$). Within the ML, Km PDE was the lowest in winter for the well-mixed stations (st 1, 13, 15), but reached its maxima at stations closer to the Rhodes cyclonic gyre (st 80, 90, 94, 111, 113) and the eastern straits (st 104–108, Table 3).

For the whole data set, PME and PDE potential rates (Vm) ranged from 0.04–18.9 and 0.017–23.4 $\mathrm{nmol\,L^{-1}\,h^{-1}}$, respectively, in winter and within a much lower range in autumn (0.014–2.7 $\mathrm{nmol\,L^{-1}\,h^{-1}}$ for Vm PME and 0.011–5.7 $\mathrm{nmol\,L^{-1}\,h^{-1}}$ for Vm PDE). The Vm of both types of phosphatases decreased with depth rapidly below the ML

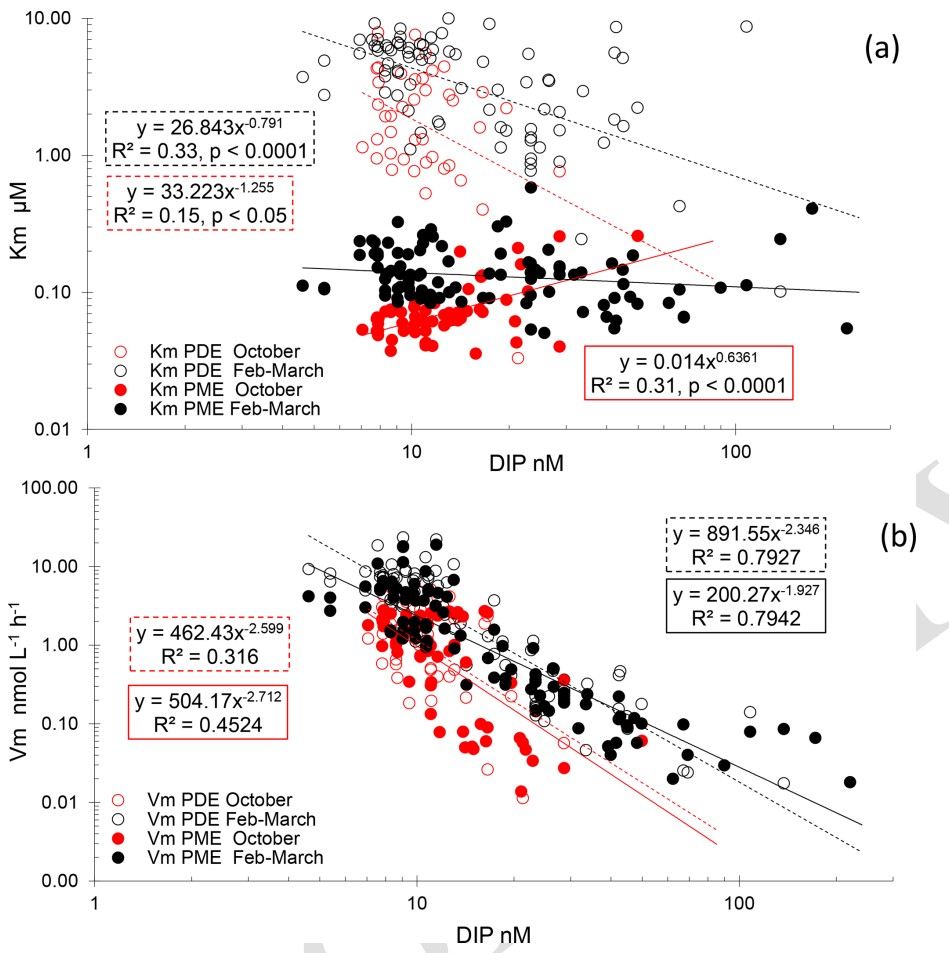

**Figure 7. (a)** Relationships between Km PME and Km PDE vs. DIP concentrations. **(b)** Relationships between Vm PME and Vm PDE vs. DIP concentrations. Open circles: PDE, full circles: PME, red: autumn cruise (PERLE1), black: winter cruise (PERLE2).

(Fig. 4d–f). Contrarily to the Km values, Vm PDE and Vm PME were on the same order of magnitude (ratio of Vm PME : Vm PDE, mean $\pm$ SD of $1.1 \pm 0.9$, range of 0.26–6.29, Fig. S2). Both rates were positively linearly corre-
lated particularly in winter, when the data range was larger (Vm PDE $= 1.38 \times$ Vm PME $+ 0.54$, $r^2 = 0.93$, $p < 0.0001$ in winter; Vm PDE $= 1.32 \times$ Vm PME $- 0.29$, $r^2 = 0.56$, $p < 0.0001$ in autumn; plots not shown). DOP and $L_{DOP}$ explained most of the lack of variability in Vm PME and
Vm PDE, whatever the cruise (log–log relationships tested). On the other hand, Vm decreased as the DIP concentration increased in all cases, with the relations being highly significant for both cruises and phosphatase types (Fig. 7b) but with lower determination coefficients in autumn (Table S1
in the Supplement). The slopes were only significantly different between the two cruises (February–March vs. October) for Vm PME ($F$ test, $p < 0.05$) and not for Vm PDE. The slope of the Vm PME–DIP relationship was only significantly lower than that of the Vm PDE–DIP relationship for
the February–March cruise ($F$ test, $p < 0.05$). Note that the log Vm $= f(\log NO_x)$ relationships were also highly signif-

icant during both cruises and for both types of phosphatase (for the four regressions, $p < 0.001$, plots not shown). In autumn, the $NO_x$ : DIP ratio was positively correlated to the ratio of Vm PME : Vm PDE (log–log regression, $r^2 = 0.40$, $p < 0.001$). On the other hand, in winter, when $NO_x$ was more available within the ML, no variability was explained in the Vm PME : Vm PDE ratio ($r^2 = 0.04$, $p > 0.05$). The turnover times (ratio of Km : Vm) of PME and PDE ranged from 0.6–257 and 11–1593 d (mean $\pm$ SE of $25 \pm 45$ and $174 \pm 317$ d), respectively.

Within the ML, means of both Vm PME and Vm PDE per station were the lowest at st 1, 13 and 15 in the Cretan Sea in winter and peaked at st 80, 90 and 94 on the easternmost transect (Table 3), revealing a great variability during the winter cruise. Within the ML, Vm was not particularly higher during the autumn cruise despite the high in situ temperature difference, and means per cruise were not statistically different: for Vm PME ($2.2 \pm 0.4 \, \mathrm{nmol \, L^{-1} \, h^{-1}}$ in autumn, $5.0 \pm 5.1 \, \mathrm{nmol \, L^{-1} \, h^{-1}}$ in winter, $p = 0.13$) and for Vm PDE ($2.9 \pm 1.3 \, \mathrm{nmol \, L^{-1} \, h^{-1}}$ in autumn, $7.6 \pm 6.8 \, \mathrm{nmol \, L^{-1} \, h^{-1}}$ in winter, $p = 0.06$).

**Table 3.** Mean PME and PDE kinetic parameters (Km, Vm) and specific Vm activities inside the ML at each station. For cell-specific activity, Vm rates are divided by the abundance of cells of Hprok; for biomass-specific Vm, Vm rates are divided by the sum of phytoplankton carbon biomass + carbon biomass of Hprok. NA: not available.

| Cruise | Station | Vm ($nmol\,L^{-1}\,h^{-1}$) | | Km ($\mu M$) | | Cell-specific Vm ($10^{-18}$ mol per cell per hour) | | Biomass-specific Vm ($nmol\,\mu g^{-1}\,C\,h^{-1}$) | |
|---|---|---|---|---|---|---|---|---|---|
| | | PME | PDE | PME | PDE | PME | PDE | PME | PDE |
| PERLE1 | 2 | 2.41 | 2.18 | 0.062 | 2.77 | 7.9 | 7.23 | 0.350 | 0.311 |
| PERLE1 | 5 | 1.49 | 1.55 | 0.060 | 2.25 | 5.1 | 5.30 | 0.227 | 0.236 |
| PERLE1 | 12 | 1.78 | 4.76 | 0.083 | 7.58 | 6.0 | 16.13 | 0.304 | 0.815 |
| PERLE1 | 15 | 2.38 | 4.00 | 0.062 | 4.14 | 6.2 | 10.48 | 0.310 | 0.521 |
| PERLE1 | 16 | 2.21 | 1.66 | 0.068 | 1.66 | 5.7 | 4.25 | 0.280 | 0.211 |
| PERLE1 | 19 | 2.68 | 4.16 | 0.068 | 4.44 | 8.3 | 12.88 | 0.356 | 0.553 |
| PERLE1 | 20 | 1.88 | 1.30 | 0.059 | 1.23 | 4.9 | 3.42 | 0.221 | 0.158 |
| PERLE1 | 23 | 2.58 | 3.70 | 0.068 | 4.03 | 7.3 | 10.46 | 0.311 | 0.456 |
| PERLE1 | 25 | 2.40 | 4.23 | 0.054 | 4.86 | 7.6 | 12.84 | 0.314 | 0.539 |
| PERLE1 | 27 | 2.71 | 1.87 | 0.075 | 1.59 | 7.9 | 5.45 | 0.378 | 0.261 |
| PERLE1 | 30 | 2.05 | 2.78 | 0.065 | 3.42 | 7.4 | 10.04 | 0.344 | 0.467 |
| PERLE2 | 1 | 0.41 | 0.26 | 0.151 | 1.62 | 1.4 | 0.83 | 0.023 | 0.014 |
| PERLE2 | 13 | 0.37 | 0.54 | 0.185 | 1.92 | 1.0 | 1.49 | 0.018 | 0.026 |
| PERLE2 | 15 | 0.22 | 0.14 | 0.209 | 1.25 | 0.6 | 0.36 | 0.013 | 0.008 |
| PERLE2 | 50 | 1.45 | 3.24 | 0.106 | 3.14 | 3.6 | 8.02 | 0.083 | 0.186 |
| PERLE2 | 104 | 4.11 | 8.54 | 0.134 | 7.05 | 10.7 | 21.93 | 0.187 | 0.392 |
| PERLE2 | 108 | 4.81 | 7.29 | 0.216 | 6.54 | 11.6 | 17.55 | 0.180 | 0.273 |
| PERLE2 | 21 | 2.61 | 3.10 | 0.118 | 1.76 | 7.7 | 9.14 | 0.139 | 0.165 |
| PERLE2 | 26 | 1.62 | 1.11 | 0.093 | 1.10 | 3.7 | 2.57 | 0.083 | 0.059 |
| PERLE2 | 35 | 3.42 | 7.91 | 0.115 | 4.20 | 10.7 | 24.78 | 0.194 | 0.448 |
| PERLE2 | 44 | 1.81 | 4.16 | 0.106 | 4.46 | 5.6 | 12.87 | 0.117 | 0.269 |
| PERLE2 | 58 | 2.84 | 4.50 | 0.136 | 4.73 | NA | NA | NA | NA |
| PERLE2 | 68 | 7.57 | 11.89 | 0.170 | 6.42 | 21.7 | 34.20 | 0.581 | 0.910 |
| PERLE2 | 75 | 4.10 | 8.72 | 0.110 | 4.31 | 14.3 | 30.38 | 0.253 | 0.537 |
| PERLE2 | 80 | 18.87 | 22.02 | 0.288 | 7.24 | 30.7 | 35.80 | 0.356 | 0.415 |
| PERLE2 | 90 | 14.53 | 20.77 | 0.260 | 6.63 | 31.9 | 45.36 | 0.539 | 0.772 |
| PERLE2 | 94 | 10.89 | 18.60 | 0.239 | 6.97 | 17.8 | 30.46 | 0.344 | 0.588 |
| PERLE2 | 111 | 4.66 | 6.71 | 0.241 | 6.61 | 6.6 | 9.46 | 0.134 | 0.193 |
| PERLE2 | 116 | 6.00 | 7.35 | 0.168 | 6.23 | 9.1 | 11.18 | 0.219 | 0.268 |

We calculated specific PME and PDE activities by normalizing over abundances of Hprok and Tchla as well as a proxy for the total living carbon biomass determined as the sum of Hprok C + phyto C (see "Material and methods"). As for Vm, all kinds of specific Vm decreased with increasing DIP concentrations, and all the relations were significant for both cruises and phosphatase types (Table S1).

Mostly, log–log regressions between Vm rates of both types of phosphatases and cell abundances of each identified phytoplankton cytometric group were significant when each population was considered individually (Tables S2 and S3). Conversely, relations were insignificant or presented a low $r^2$ ($< 0.27$) for heterotrophs (HNA and LNA total Hprok or HNF). To avoid autocorrelations between variables (Vm rates and abundances of all cytometric groups tended to decrease with depth) we also examined partial correlations coefficients using multiple log–log regressions, using all cytometric groups as independent variables. Cryptophyte-like cells and *Synechococcus* were the two populations explaining variability in Vm PME during the PERLE2 cruise, whereas it was picophytoeukaryotes and *Synechococcus* during the PERLE1 cruise (Table S2). For Vm PDE, it was the same two populations for the PERLE2 cruise as for Vm PME, expect that it was only *Synechococcus* for the PERLE1 cruise (Table S3). Abundances of LNA cells, HNA cells and HNF cells never significantly explained any variability in Vm rates in the multiple regressions.

## 4 Discussion

### 4.1 Phosphate pools and P stress

During both cruises, DIP showed classical nutrient profiles with depleted DIP within the surface layers and increasing

values at depth. DIP stocks showed low values within the upper layers, with means in the mixed layer varying from 5 to 24 nM according to the station. Over a larger spatial scale in the Mediterranean Sea, Pulido-Villena et al. (2021) obtained 6–15 nM in spring in the phosphate-depleted layer across the Ionian and western basins, and over a large timescale, values were reported to be $\sim 6$ nM throughout the entire year in the Levantine Sea (Ben-Ezra et al., 2021).

$L_{DOP}$, the pool of DOP hydrolysable by a phosphomonoesterase purified from *E. coli*, was lower than 25 nM, except for a few samples, and was in the same range as in above-nutricline waters of the central North Pacific (from detection limits to 40 nM; Yamaguchi et al., 2019). The $L_{DOP}$ depth profile pattern did not matched that of DIP: $L_{DOP}$ displayed constant values with depth with no particular peak within the DCM. A shift towards a nutrient-like distribution has been reported only in some of the coastal stations examined by Hashihama et al. (2013), who suggested these were under severe P stress. On the other hand, Yamaguchi et al. (2019) showed constant profiles or occasionally higher $L_{DOP}$ peaks within the 0–100 m layer within the low- to middle-latitude central North Pacific, along a transect including stations under moderate P stress. $L_{DOP}$ concentrations were shown to be lower at the basin scale under low-DIP ($< 100$ nM) conditions when compared to $> 100$ nM conditions in the moderate P-stressed area explored by Yamaguchi et al. (2019) in the Pacific. Conversely, in our study, there was no significant linear correlation between $L_{DOP}$ and DIP (Fig. S3), but our DIP concentrations varied on a lower range (mean $\pm$ SD of $36 \pm 48$ nM), and all the data in the ML were below 26 nM.

The fraction of $L_{DOP}$ in DOP (%$L_{DOP}$) varied within a large range, from 1.5 % to 97 %, with a mean of 28 % $\pm$ 18 %. This mean is in the same range as in Djaoudi et al. (2018a) (27 $\pm$ 19 %) in a year survey of epipelagic layers in the western Mediterranean Sea (ANTARES offshore station). It was on average lower (7 $\pm$ 5 %) in a moderate P-stress oceanic area (Pacific; Yamaguchi et al., 2019) but was variable along a salinity gradient in the DIP-rich (0.3–1.9 µM DIP) Tamar estuary (0.7 to 79 %, with a mean of 35 % $\pm$ 21 %; Monbet et al., 2009).

$L_{DOP}$ accounted for a large and variable percentage of the DOP pool, suggesting that other components of the DOP might play a role in P cycling. The variability in the Vm PDE and Km PDE estimated in our study suggests that P-diesters could be an important P source for marine microorganisms. Marine P-diesters, like P monoesters, have been quantified based on hydrolysis of DOP by purified enzymes (Suzumura et al., 1998; Monbet et al., 2009; Yamaguchi et al., 2019). Although no data have been quantified to date in the Mediterranean Sea, other measurements suggest that P-diesters could represent as much as P monoesters. In a nutrient-rich estuary (DIP ranged from 0.28 to 1.2 µM) under strong salinity gradients and interaction with sediment porewaters, P-diesters contributed on average 29 % of the DOP compared to 35 % for P monoesters (Monbet et al., 2009). Along 170° N in the Pacific Ocean, marine P-diesters in epipelagic layers (on average $5 \pm 6$ nM) are lower than P monoesters (on average $12 \pm 5$ nM) (Yamaguchi et al., 2019). P-diesters include a large panel of molecules as nucleic acids, nucleotides or phospholipids. Among these forms, phospholipids' P concentration ranged from 0.6–25 nM in coastal areas (Suzumura and Ingall., 2001). In Goutx et al. (2009), dissolved phospholipids in Mediterranean Sea amounted to on average around 1 µg C L$^{-1}$ and up to 3.7 µg C L$^{-1}$. Based on an average carbon chain length of 16 for fatty acid, P would represent around 6.8 % of the phospholipid carbon mass; i.e., dissolved phospholipids would be around 2.2 nM P, up to 8 nM, which could be considered minimal ranges of P-diester concentrations, as they include also other P-diesters types. Thus, both P monoesters and P-diesters should be considered in the P cycle in the Mediterranean Sea.

## 4.2 Phosphatase kinetics

Our study is the first one describing simultaneously PME and PDE activities in the Mediterranean Sea. Furthermore, to our knowledge, this is the first study systematically describing Michaelis–Menten equations for PDE until the saturation state. Indeed, both types of phosphatases displayed typical Michaelis–Menten kinetics, but PME saturated after the addition of 1 µM MUF-P and PDE saturated only after the addition of 50 µM bis-MUF-P. Kinetic parameters Vm and Km are thus difficult to compare with previous literature, as Km and Vm depend on the range of the concentration of fluorogenic substrates added, with recommendations to add up to 10 times the Km value to calculate Vm appropriately (Urvoy et al., 2020). In most cases only one single substrate concentration is used: for instance, Sato et al. (2013) compared PME and PDE rates using 1 µM substrate concentration, Thomson et al. (2020) used 100 µM concentrations, and Huang et al. (2022) used 1 mM concentrations. In addition, while some authors used MUF derivatives (Sato et al., 2013; Thomson et al., 2020), others used paranitrophenyl derivatives (Huang et al., 2022), probably corresponding to different enzyme affinity. In addition, conditions of incubation may also differ, with some authors using in situ or nearly in situ temperatures (Sato et al., 2013; Suzumura et al., 2012; Yamaguchi et al., 2019; Thomson et al., 2020) and others using optimal temperatures (Huang et al., 2022).

Sato et al. (2013) and Suzumura et al. (2012) explored PME activities at 10 m and at the DCM depth in the North and South Pacific, where DIP concentrations varied from 3 nM to hundreds of nanomolars. They did some kinetics with PME up to 1 µM MUF-P concentrations, and thus their results are comparable to ours for this enzyme: they found maximum PME hydrolysis rates (Vm) reaching at best 3.7 nmol L$^{-1}$ h$^{-1}$. In our study we obtained values up to 18 nmol L$^{-1}$ h$^{-1}$, confirming the other high values already obtained in the western and central Mediterranean Sea in

spring or in winter in the open sea (up to $13 \, \text{nmol L}^{-1} \text{h}^{-1}$, Van Wambeke et al., 2002, 2021). Vm PDE seems to range also in the same order of magnitude as Vm PME. Thomson et al. (2020) measured PDE and PME potential rates at $100 \, \mu\text{M}$ concentrations, at a station in the South Pacific located 65 km off the Otago coast, in sub-Antarctic waters. At the surface (2 m) and 500 and 1000 m depths explored in their study, DIP was always detectable ($0.5$–$1 \, \mu\text{M}$). The 2 m depth layer presented seasonal variations, with PME rates varying from $2.7$–$12 \, \text{nmol L}^{-1} \text{h}^{-1}$ and PDE rates from $1.4$–$20 \, \text{nmol L}^{-1} \text{h}^{-1}$. The comparison of this study with others previously assessing PME and PDE activity rates (Sato et al., 2013; Thomson et al., 2020) reveals similar patterns. Indeed, in all cases, Vm was on the same order of magnitude for both phosphatase enzymes and its variability was better explained by DIP than by DOP or L$_{\text{DOP}}$. This similarity of patterns among oceanic regions occurs despite contrasting environmental conditions. The sub-Antarctic waters sampled by Thomson et al. (2020) are located in a high-nutrient, low-chlorophyll (HLNC) region rich in macronutrients (DIP ranged from $0.5$–$18 \, \mu\text{M}$) and poor in trace metals. On the other hand, part of the region covered by Sato et al. (2013) (the northwestern Pacific) is not iron-limited but P-limited (Liang et al., 2022), similar to the eastern Mediterranean Sea (Statham and Hart, 2005; Thingstad et al., 2005), although phytoplankton in the eastern Mediterranean Sea can be N + P co-limited and heterotrophic prokaryotes can be labile C + P co-limited (Van Wambeke et al., 2002; Thingstad et al., 2005; Tanaka et al., 2011). In the North Pacific, nitrogen fixation occurs and is mainly expressed by cyanobacterial diazotrophs like *Trichodesmium* and *Crocosphaera* (Horii et al., 2023). In the eastern Mediterranean Sea, dinitrogen fixation represents a small contribution to primary production (Rahav et al., 2013) and is expressed essentially by heterotrophic prokaryotes. Further, these heterotrophs are more controlled by organic C availability than by iron (Sisma Ventura and Rahav, 2019). Finally, it is in the eastern Mediterranean Sea that lower DIP turnover times have been measured ($< 10$ h, Talarmin et al., 2015) compared to the southwestern Pacific (10–100 h, Van Wambeke et al., 2018) or the North Pacific (48–939 h, Sohm and Capone, 2010, and references therein).

In Thomson et al. (2020), the ratio of PME : PDE rates at the surface was lower or higher than 1 (range of 0.5 to 5.3) and varied seasonally. Their surface values were negatively linearly correlated with DIP and positively with the NO$_x$ : DIP ratio. In our study, the NO$_x$ : DIP ratio in the ML was on average much lower in autumn than in winter ($1.8 \pm 0.7$ compared to $55 \pm 19$), traducing a possibly different degree of limitation (N + P co-limitation in autumn, P limitation in winter). Over our whole data set, the Vm PME : Vm PDE ratio was not related to DIP or to the NO$_x$ : DIP ratio. Only under autumn conditions was a positive correlation observed between the Vm PME : Vm PDE ratio and the NO$_x$ : DIP ratio, similar to what was observed by Thomson et al. (2020). Intuitively, it is expected that heterotrophic bacterial communities and/or some phytoplankton groups would develop more nucleotidases, deoxyribonucleases (DNases) or ribonucleases (RNases) relative to monoesterases when NO$_x$ also becomes limiting in regard to DIP, as such types of PDEs allow for simultaneous access to both organic P and N sources. As in our study the correlation between Vm PME : Vm PDE and NO$_x$ : DIP in autumn was estimated including all data; this relationship must consider also NO$_x$ : DIP changes along the vertical column. Indeed, NO$_x$ : DIP ratio increased within the DCM layer, associated with higher Vm PME : Vm PDE ratios. Within the DCM, besides lower light, the N source is more energetically available (i.e., reduced) due to the nitrification process, and nitrate that is more available as Ncline is shallower than the Pcline. Moreover, Thomson et al. (2020) also suggested that the variability in Vm PME : Vm PDE could traduce shifts in communities expressing different genes or in the availability of different P esters. As a consequence of being recognized as a different biogeochemical niche, the DCM layer also presents different communities of phytoplankton and heterotrophic bacteria than the ML layers (Scharek and Latasa, 2007; Dupont et al., 2015; Estrada et al., 2016; Crombet et al., 2011), i.e., different populations possibly having different types of genes expressing phosphatase activities. During the winter cruise, multiple regression revealed that *Synechococcus* and cryptophyte-like cells mostly explained the variability in Vm PME and Vm PDE. Further, we probably could not accurately determine the abundances of *Prochlorococcus* cells by flow cytometry despite the special setting of the machine used to specifically enhance the detection of this population having very dim fluorescence at the surface, particularly in autumn, when divinyl chlorophyll *a* was above the limits of detection in the mixed layer. This is a very common feature already described in the literature (Mella-Flores et al., 2011; Reich et al., 2022). Based only on multiple regression analysis, it is difficult to establish a causal link between phytoplankton groups and phosphatase activities.

Sato et al. (2013) and Suzumura et al. (2012) found Km PME ranging from 0.08 to $1.3 \, \mu\text{M}$ in the Pacific Ocean. Yamaguchi et al. (2019) used more systematically MUF-P concentration kinetics of up to $2 \, \mu\text{M}$ along vertical profiles in the North Pacific and obtained Km ranging from 0.095 to $1.9 \, \mu\text{M}$. Data in our study were in the lower range of the above-cited publications. It is however possible that we achieved higher sensibility for Km determination as we performed concentration kinetics with six concentrations starting at $0.025 \, \mu\text{M}$ MUF-P. Sato et al. (2013), from their small data set (10 kinetics), observed a positive relationship between Km PME and DIP, which was considered an environmental adaptation through production of ectoenzymes with a higher affinity for the substrate (i.e., low Km) when the degree of P deficiency increases (i.e., low DIP). In this study, the strongly stratified conditions in the post-bloom case within the Ierapetra gyre effectively led to a lowering of Km PME and of L$_{\text{DOP}}$ inside the ML compared to the

winter cruise (Tables 2 and 3) and thus a better affinity for the substrate for PME in autumn when $L_{DOP}$ is low. Going further with the data set per cruise over the water column, we found either no (February–March cruise) or a positive relationship (October cruise, Fig. 7a) between Km PME and DIP concentrations, showing that Km does not always follow the Sato et al. (2013) concept. Furthermore, a consistent fact in comparison with the few authors that used MUF-P concentration kinetics simultaneously with measurements of the different DOP pools is that Km PME was related neither to DOP nor to $L_{DOP}$ concentrations, i.e., confirming that the DIP is the driving force for PME activity, not the enzyme substrate source. As previously discussed, it is possible that $L_{DOP}$ does not reflect the real conditions of accessibility to the substrate pool (Duhamel et al., 2011; Suzumura et al., 2012). Indeed, why would microorganisms express enzymes having kinetic properties with Km PME being about 13-fold higher than $L_{DOP}$ stocks? Possibly intermittent sources and the patchiness of $L_{DOP}$ composition and concentration could explain a high Km value relative to $L_{DOP}$ so that microorganisms maximize their PME activities at high $L_{DOP}$ concentrations. Patchiness is the consequence of the size continuum of organic matter with different molecular composition from low molecular weight to high molecular weight (Young and Ingall, 2010). Patchiness is provoked, for instance, during the passage of sedimenting particles with their associated plumes (Kiørboe et al., 2001), phases of intense lysis of cells, egestion of food vacuoles by grazers (Nagata and Kirchman, 1992) or hydrolysis of particulate detritus. In addition, since PME mostly originates intracellularly or from the periplasm of cells (Luo et al., 2009), it is probably adapted to higher concentrations of DOP than that estimated by the bulk DOP measurement. Other caveats could be connected to the representativity of an artificial fluorogenic substrate or the difficulty in assessing PME activity under concentrations of a few nanomolars of MUF-P (Pulido-Villena et al., 2021).

For Km PDE, Sato et al. (2013) found no correlation with DIP, whereas we found a negative one, observed in winter as well as in autumn (Fig. 6a); i.e., the affinity for the PDE substrate increases when DIP increases, which is counterintuitive as long as we consider DIP as a proxy for P limitation and as long as we consider that PDE is induced under P stress. Such negative correlations could be related to the depth effect: indeed, Km PDE tended to decrease with depth, and this was not the case for the Km PME. Thus, the variability in Km PDE with depth could be connected to probable changes over the epipelagic layers in nutrient stress (P vs. N + P), in the composition of natural PDE substrates and/or in the presence of different types of PDE according the consortium of microorganisms present, as discussed above for the Vm PME : Vm PDE ratio.

## 4.3 Bioavailability of organic P

As a consequence of the great difference in Km, the turnover time (TT) of the P-diester pool was ∼ 7 times higher than for P monoesters (TT means of 26 d for PME and 175 d for PDE). PDE and PME turnover times are difficult to compare considering the different range of concentrations used in the other field studies, but most authors agree on a higher TT for PDE. In Sato et al. (2013) TT PME ranged from 5–112 d, i.e., about 1 order of magnitude lower than for PDE (128–535 d), with concentration kinetics of up to 1 µM fluorogenic substrate. In Yamaguchi et al. (2019), expanding their data in the Pacific down to the Equator, mean TTs were $99 \pm 75$ d for PME and $2944 \pm 1224$ d for PDE, with concentration kinetics using up to 2 µM fluorogenic substrate. A higher PDE turnover time suggests that P-diesters are a slowly degradable fraction of the DOP. However, P-diesters include a large panel of molecules which might have different turnover times based on their chemical nature and solubility. A methodological bias explaining a high PDE TT is that the substrate used, bis-MUF-P, does not seem to be efficiently hydrolyzed using a purified PDE type I from venom, whereas other artificial P-diester substrates such as p-nitrophenyl thymidine 5'-monophosphate (pNP-TMP) are hydrolyzed under the same conditions (Yamaguchi et al., 2019). The inconvenience is that the hydrolysis of pNP-TMP is followed by colorimetry, resulting in much less sensitivity than with bis-MUF-P, for which hydrolysis is followed by fluorimetry and which does not allow for running concentration kinetics with very low concentration of substrates. The bias also seems to be present using another artificial substrate, as bis-paranitrophenyl phosphate was only partly hydrolyzed in conditions where DNA was almost fully hydrolyzed (Monbet et al., 2007; Turner et al., 2002).

The high-molecular-weight (HMW) fraction of the DOP was submitted to enzyme digestion by purified PME and PDE in coastal seawater off Tokyo Bay (Suzumura et al., 1998). In this study, the HMW fraction contained 1/3 of the total DOP pool and 5 times more P-diesters than P monoesters (7 % P monoesters, 48 % P-diesters and 44 % non-reactive DOP), which confirmed an unequal distribution of P-diesters compared to P monoesters. Accessibility to the P-diesters for the enzymes also plays a role in its degradability as P-diesters might be embedded in HMW fractions such as colloids, virus-like particles, vesicles or submicron particles (Biller et al., 2022) so that P-diesters could be not accessible to the purified enzyme during the assay, as discussed in Suzumura et al. (1998) and Monbet et al. (2007). Possibly, the localization of natural enzymes also differs along the size continuum of organic matter as has been shown for bacterial phosphatases (Luo et al., 2009), although to date PME and PDE activities were equally distributed based on studies using size fractionation: Thomson et al. (2020) found mostly equal (87 %–88 %), cell-free (< 0.2 µm) proportions of PME and PDE activities in cold, sub-Antarctic waters

(87 %–88 %), and Huang et al. (2022) found mostly high proportions in the nanometer–micrometer ($> 2\,\mu$m) size fraction for both enzymes ($> 74$ %) in a temperate, rich coastal area under bloom conditions. The long turnover times of PDE obtained in our study, whatever the season, stratification conditions, and N or N + P degree of limitation, suggest that P-diesters are more stable than P monoesters, although care should be taken to determine the accessibility to the substrate by the enzyme and the representativity of analog substrates in future studies.

## 5 Conclusion

This is the first study showing the distribution of both phosphomonoesterase and phosphodiesterase in the Mediterranean Sea, via systematic use of concentration kinetics. This approach avoids biases connected to the use of a single concentration or range of concentration not adapted to Vm PDE estimates. This study confirmed the general trend obtained in other studies, i.e., that Vm PDE and PME seem to be more controlled by DIP availability rather than by the substrate availability. Although DIP concentration remained more or less constant within the surface mixed and DIP-depleted layer, the large changes in Vm and percentages of $L_{DOP}$ obtained according to the station and season suggest strong adaptations of microbial populations and large degrees of P limitation. The much higher Km values and turnover times obtained for PDE compared to PME suggest different accessibility to the substrate P monoesters and P-diesters along the size continuum of organic matter. Opposite changes in the kinetic parameters of PDE and PME (Km values, Vm PME : Vm PDE ratio) with depth suggest adaptations of the microorganisms producing them along the epipelagic layer as they are submitted to different biogeochemical forcings. To better characterize such microbial adaptations to P deficiency in future studies, a necessary approach is to combine biogeochemistry with microorganisms' physiology, for example by simultaneously following gene expression of phosphatase families, determining the composition of DOP along the size continuum of organic matter and measuring in situ hydrolysis rates of different types of P-containing organic molecules.

*Data availability.* Data collected by the two oceanographic cruises are available at the operational oceanographic data center Coriolis (https://doi.org/10.17600/18000865, Durrieu de Madron and Conan, 2019).

*Supplement.* The supplement related to this article is available online at: https://doi.org/10.5194/bg-21-1-2024-supplement.

*Author contributions.* FVW and EPV conducted the experiment, analyzed PME and PDE, and wrote the first draft of the manuscript. MPP, OC and AP analyzed nutrients. VT provided maps and discussed mesoscale variability. MD provided the analysis of phytoplankton by flow cytometry. CS provided the analysis of heterotrophic prokaryotes and nanoflagellates by flow cytometry. PC and MPP reviewed and edited the manuscript. All authors contributed to the article and approved the submitted version.

*Competing interests.* The contact author has declared that none of the authors has any competing interests.

ther geographical representation in this paper. While Copernicus Publications makes every effort to include appropriate place names, the final responsibility lies with the authors.

*Acknowledgements.* We warmly thank many people for their help on board: Frank Dumas (chief scientist of the PERLE1 cruise); Florian Voron (for DIP analysis); Sophie Guasco and Thibaut Wagener (for help with PDE and PME measurements); Catherine Guigue (for phospholipid data and P conversion); Paul Labatut, Barbara Marie and Eric Maria (for nutrients and the sampling of dissolved organic material as well as ammonium analyses); and Fabrizio D'Ortenzio and Joelle Salaun (for pigment sampling). We thank the services provided by the SAPIGH (https://lov.imev-mer.fr/web/facilities/sapigh/, last access: 5 May 2024), BioPIC (https://www.obs-banyuls.fr/fr/rechercher/plateformes/biopic.html, last access: 5 May 2024) and PRECYM (https://precym.mio.osupytheas.fr/, last access: 5 May 2024) platforms for chlorophyll and flow cytometry analyses.

*Financial support.* This study is a contribution of the PERLE project, a joint initiative of the Chantier Méditerranée MERMeX supported by CNRS INSU, Ifremer, CEA and Météo-France as part of the MISTRALS program coordinated by INSU. PERLE1 (PROTEVS cruise) was also partly managed and founded by the Service Hydrographique et Océanographique de la Marine (SHOM; Brest, France) (funded by the French Direction générale de l'armement, DGA). Pigment analysis was paid for by the Equipex NAOS (Novel Argo Ocean observing System) program.

*Review statement.* This paper was edited by Perran Cook and reviewed by two anonymous referees.

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
