# Peer review of "Phosphomonoesterase and phosphodiesterase activities in the Eastern Mediterranean in two contrasted seasonal situations"

_EGUsphere, 2023_

## Author Comment (AC1)

**RC1**: 'Comment on egusphere-2023-2578', Anonymous Referee #1, 04 Dec 2023 reply

General Comments

This article reveals the distribution of phosphatemonoesterase (PME) and phosphatediesterase (PDE) activities in the seawater of the Mediterranean Sea in the fall and winter. Primary production in the survey area is known to be limited by low phosphors bioavailability, and the magnitude of the limitation varies seasonally, in accordance with a variation in the stratification intensity there. The activities of these two enzymes are indices of microbial phosphorus stress and they provide bioavailable inorganic phosphate to microbes via hydrolysis of organic phosphorus materials. The scientific significance of this study is obvious, and the validity of the methodology employed in this study is thoroughly examined in this article. The results and discussion provided in this article evidently contribute to understanding the biogeochemical cycling of phosphorus and relevant biogenic materials in this area.

**Response:** We appreciate that the reviewer has found the work interesting and worthwhile to consider for publication. We truly thank him/her for providing detailed and useful comments, and we have addressed carefully his/her feedback.

Specific responses to the reviewers' comments are provided below in blue, modified sentences included in the revised version in blue + italics and line numbers referring to the new revised version with track changes are highlighted in yellow.

One of the inputs that I can make to this article is a comparison with the other phosphorus limiting areas. Particularly, the western North Pacific is also known to be heavily limited by phosphorus deficiency, and PME and PDE activities have been already reported from it. Despite these similarities between the Mediterranean Sea and the western North Pacific, they are different in many respects, including trace metal supply (especially iron and copper), nitrogen fixation (high abundance of *Trichodesmium* and *Crocosphaera* in the North Pacific), and microbial community composition. The comparison should be of great significance to contrast the difference between the two areas, and possibly with other phosphorus-limiting water(s).

**Response:** We fully agree with this comment and we have added a discussion about the characteristics of other phosphorus-limited areas (see below).

In addition, I recommend the authors to incorporate a discussion on the role of meso- and microscale eddies observed during the campaign, particularly in the winter season. The eddy structure can profoundly influence the vertical structure of the water column, as already described in the article, and hence the microbial community structure and their biological processes. Despite its experimental design, I do not think that the article sufficiently discusses this viewpoint. I believe that these insights can improve the quality of this article.

**Response:** We deliberately focused this manuscript on the variability of PMA and PDE kinetics linked to methodological issues and to seasonal aspects. We fully agree with the reviewer that the experimental design of the cruise offered a nice occasion to assess the role of mesoscale structures and this is developed in a companion paper which is almost ready for submission:

*'Van Wambeke et al., Mesoscale variability of phosphorus stocks and biogeochemical fluxes in the mixed layer during 2 contrasted seasons in the Eastern Mediterranean Sea'. This second manuscript is now cited at the end of the introduction* (line 105-108) *as follows: 'A second paper in preparation (Van Wambeke et al., in prep) will be dedicated to PDE and PME distribution within the surface mixed layer in relation to mesoscale variability (cyclones vs anticyclones) and the progression of the phytoplankton bloom in winter.'*

We can provide a draft to the referee upon request.

Finally, we propose to change the title of our article as: *'Phosphomonoesterase and phosphodiesterase activities in the Eastern Mediterranean Sea in two contrasted seasonal situations'*

Specific Comments

L62 Describing typical P-diester chemical species found in seawater will be useful for the upcoming discussion.

The following sentence was added line 67-71:*' In aquatic environments, typical P-diesters identified are nucleotides, nucleic acids, and phospholipids coming from microorganism's intracellular material (Karl and Bjorkman, 2015), but the methodology used to estimate the P-diester pool (using also a commercially purified phosphodiesterase enzyme (Monbet et al., 2007; Yamaguchi et al., 2019)) does not allow to determine the in-situ P-diesters chemical composition in detail.'*

L115 Here the authors said that they obtained "pigment distributions", but only the distribution of total chlorophyll *a* was provided in this article. Describe more accurately or provide all the results of pigment analyses and discuss them.

We agree. The term 'pigment distribution' (line 127) has been removed and replaced by the term 'chlorophyll a'

L125 What do the figures with a ± sign mean?

These values present the analytical precision of the method. To avoid confusion, this was rephrased line 138-141 as: *'Micromolar nutrient concentrations of nitrate, nitrite and phosphate were determined by colorimetry (Aminot and Kérouel, 2007) using a segmented flow analyzer Seal-Bran-Luebbe (AAIII HR SealAnalytical©), with analytical precision of 0.02 μM, 0.01 μM and 0.01 μM, respectively.'*

L141 The species name *coli* should be written in lower case.

This has been done

L169 Chlorophyll should be written in lower case.

This has been done. We checked all the abbreviations written as 'Tchla' in the text and figures

L185 Does this "PRISM" mean a statistic software? Then describe its properties more in detail.

Yes, it is a statistical software. Its internet link has been added in the revised version as follows (line 228): *'…errors were estimated by non-linear regression (software PRISM https://www.graphpad.com/features) using the Michaelis-Menten equation…..'*

L256 Out of these two figures, the former should be from the autumn cruise, but it is not clear from the text. Make clearer which figure is which.

We are not sure of which figure the referee is referring to. Dealing on Fig. S3, both plots a) and b) presents October data (in red) and Feb-March data (in black). Both figures have $L_{DOP}$ in Y axis, so on both of them it is possible to see that in winter, only few data of $L_{DOP}$ are higher than 25 nM. Stations are however not separated on Fig. S3. On Fig. 4, it is possible to look at more precisely $L_{DOP}$ data of station 75 (Fig. 4c) and stations 1 13 15 (Fig 4b) in which some values were > 25 nM.

In case the referee talk about 'numbers' cited line 256 and not about 'plots', the last sentence of this paragraph was also modified as: *For the whole data set, the fraction of $L_{DOP}$ in DOP (%$L_{DOP}$) varied on a large range, from 1.3% to 97%, with a mean of 28% ± 18%.*

L255 The sequence of the supplementary figure numbers does not match with that of appearance in the text. Correct it.

If we are not wrong, Fig S1 is cited first (line 206), then Fig. S2 (line 250) then S3 (line 255), S4 (line 276), S5 (line 277) and S6 (line 279) (line numbers of the version submitted).

L274 More abundant than what?

Sorry, more abundant than in winter. The sentence has been corrected accordingly line 319: *'In autumn, all picophytoplankton groups were more abundant than in winter'*

L280 Including "Nanoeuk" and "crypto" into picophytoplankton groups is not natural.

Yes, sorry for that. The sentence has been corrected lines 324-325 as follows: *'In winter, following mixing/stratification conditions, all phytoplankton groups (Syn, Proc Picoeuk, Nanoeuk, crypto) as well as Hprok were low and relatively homogeneous….'*

L295 The authors use the word "than" here, but I am not sure what and what are compared here.

The sentence was reformulated (line 342-344) as: *'After testing a large set of substrate concentration between 25 nM and 50 µM on some samples, the saturation state was reached at different concentrations for PME and PDE (Fig. 5).'*

L298 Is this figure "33 ± 25" the arithmetic average of all the ratios of PME and PDE in the same subsamples with a standard deviation?

Yes, we calculated the ratios KmPDE:KmPME for the whole data set (i.e., a ratio for each sample, including both winter and autumn samples) and then calculated the mean of all these ratios, and the standard deviation corresponding to this mean. To avoid confusion, the sentence was reformulated (line 347) as: *'On average for the whole data set, the Km PDE was 33-fold higher than that of the PME (mean ± sd : 33 ± 25), however…'*

L319 Are these rates correlated positively or negatively? It would be more informative to make it clear about that.

Correlation are positive, this adverb was added in the sentence line 371. Note that even if plots are not shown, the equations are given after the sentence.

L325 The statistical test for a difference in slopes of correlation curves is usually done by $F$-test. Additionally, 0.03 is uncommon for the criterion of the level of significance.

Although comparison between slopes can be done using both F- and t-tests (Andrade and Estévez-Pérez, 2014), in the revised version of the manuscript, we now assess the difference between slopes using F-test. The results remain the same. Also, we now specify the level of significance using $p < 0.05$ (line 377).

L336 Do the authors mean $V_m$ by "maximum rates"?

Yes. To avoid confusion, we now use the term 'Vm' in the revised version (line 388)

L359 As I mentioned in the General Comments, add the discussion from the more detailed comparison of the present results with those from the other phosphorus limiting areas, with special emphasis on the differences in environmental conditions, including physical, chemical, and biological conditions. In addition to such comparisons, the effects of eddy structures on microbial processes of phosphorus cycling in this area should be examined.

As suggested by the referee we added a comparison of the present results with those from the other phosphorus limiting areas, with special emphasis on the differences in environmental conditions. Some sentences were added lines 497-514 as follows: *'The comparison of this study with others previously assessing PME and PDE activity rates (Sato et al. 2013, Thomson et al. 2020) reveals similar patterns. Indeed, in all cases, Vm was on the same order of magnitude for both phosphatase enzymes and their variability was better explained by DIP than by DOP or LDOP. This similarity of patterns among oceanic regions occurs despite contrasting environmental conditions. The Subantarctic waters sampled by Thomson et al. (2020) are located in a HLNC region rich in macronutrients (DIP ranged 0.5-18 µM) and poor in trace metals. At the opposite, part of the region covered by Sato et al. (2013) (the North West Pacific) is not iron-limited but P-limited (Liang et al., 2022), similarly to the Eastern Mediterranean Sea (Statham and Hart, 2005, Thingstad et al. 2005), although phytoplankton in the eastern MS can be N+P co-limited and heterotrophic prokaryotes labile C+P co-limited (Van Wambeke et al., 2002; Thingstad et al., 2005; Tanaka et al., 2011). In the North Pacific, nitrogen fixation occurs and is mainly expressed by cyanobacterial diazotrophs like Trichodesmium and Crocosphaea (Horii et al., 2023). In the eastern MS, dinitrogen fixation represents a small contribution to primary production (Rahav et al., 2013) and is expressed essentially by heterotrophic prokaryotes. Further, these heterotrophs are rather controlled by organic C availability than by iron (Sisma-Ventura et al., 2019). Finally, it is in the eastern MS and in the Sargasso Sea that the lowest DIP turnovertimes have been measured (< 10 h) compared to the South West Pacific (10-100 h) or the North Pacific (90-1000 h, Moutin et al., 2008 and references therein).*

As explained before, we did not develop address the effect of eddy structures in this manuscript as it is the main focus of a companion paper in preparation.

L377 "always largely" sounds self-contradictory. Which is correct, always (100%) or largely (>~80%)?

It is always, see the modified sentence below.

L378 What do the authors mean by "best"? What is "good" here (same for L416)

The sentence was reformulated lines 434-436 as: *'Conversely, in our study, there was no significant linear correlation between $L_{DOP}$ and DIP (Fig. S3), but our DIP concentration varied on a lower range (mean ± sd 36 ±48 nM), and all the data in the ML were below 26 nM.'*

L424 Does it mean "From studies where both phosphatase rates were available"?

Yes, thanks, we corrected the word accordingly: *'From studies where both phosphatases…'*

L432 Relationship between what and what? L432 What do the authors compare here with the ratio $V_{mPME}$:$V_{mPDE}$?

We reformulated the sentence to avoid confusion lines 525-528 as: *'Over our whole data set, the ratio Vm PME:Vm PDE was not related to DIP or to the ratio NOx:DIP. Only under autumn conditions a positive correlation was observed between the ratio Vm PME:Vm PDE and the ratio NOx:DIP, similarly to what was observed by Thomson et al. (2020).'*

L439 What do the authors mean by "effectively"? What is more effective to what?

The sentence was reformulated (lines 534) as: *'Indeed, NOx:DIP ratio increased within the DCM layer, associated to higher Vm PME:Vm PDE ratios '*

L448 "In autumn cyanobacteria switched from…" "Switch" suggests a temporal change from one state to another, but in this situation, the authors only describe a vertical (spatial) variation of a plankton community. It sounds misleading.

This part of the discussion was reformulated, see below

L455 Here the authors' speculation is unclear from the text. Rearrange the discussion line and make it clear what is suggested from the present results.

This part of the discussion was reformulated, lines 542-571 as follows:*'During the winter cruise, multiple regression revealed that Synechococcus and Cryptophyte-like cells explained most of the variability of Vm PME and Vm PDE. However, we probably did not determine accurately the abundances of Prochlorococcus cells by flow cytometry despite the special setting of the machine used to specifically enhance the detection of this population having very dim fluorescence in surface, particularly in autumn, when dv-chla was above the limits of detection in the mixed layer. This is a very common feature already described in the literature (Mella-Flores et al., 2011; Reich et al., 2022). Therefore, based only on multiple regression analysis, it is difficult to establish a causal link between phytoplankton groups and phosphatase activities.*

L528 Turnover times should be evaluated by "long" or "short". The description "high turnover times" is ambivalent whether the turnover is fast or slow.

We changed the term (line 657) 'very high' by 'long'

Tables. Is there any authors' intention in providing the results from the cruise PERLE2 in the upper rows, although this cruise was conducted later and in the figures, the results from this cruise are provided in the lower panels? It may be confusing.

We modified Table 1, 2, 3, S2, S3, providing PERLE1 results first.

Fig. S4a. The near-zero abundance of *Prochlorococcu*s in the surface water is not realistic considering the physical and chemical conditions of the water. I am afraid that very faint cellular autofluorescence of *Prochlorococcus* within the well-lit stratified water layer resulted in failure to detect them by flow cytometry. If any other data based on genomic analyses or microscopy are available, I recommend the authors to check the validity of these data. Otherwise, the authors can show some caveats in the materials and methods section. Anyway, it is a frequently seen situation, and it will not critically affect the discussion in the present study.

We agree with this comment. Commercially available flow cytometers do not always completely resolve populations from background noise due to the low fluorescence of surface *Prochlorococcus* cells and this has been claimed since a long time ago (Partensky et al., 1999), and also in the eastern Mediterranean Sea (Garczareck et al., 2007; Mella-flores et al., 2011). The flow cytometer used was a Facs Calibur (BD) set on 2 different protocols: one for nano and pico phytoplankton and one specifically designed for *Prochlorococcus*, the difference lies on the amplification of the photodetector of red fluorescence signal (873 UA vs 608). We added more precision in the M&M section (lines 189-199) as:

*'Phytoplankton Samples were analyzed according to Marie et al. (2000) protocols using the FACSCalibur (BD Biosciences ®) of the PRECYM flow cytometry platform (https://precym.mio.osupytheas.fr/), equipped with a blue (488 nm) laser and a red (634 nm) laser. Just before phytoplankton analyses, 2 µm beads were added as an internal standard and to discriminate picoplankton (< 2-3 µm) and nanoplankton (> 2-3 µm) populations (Fluoresbrite YG, Polyscience). A Trucount beads (BD Biosciences ®) solution was also added to the samples to determine the volume analysed. The same sample was acquired twice using two different settings: the first one to assess picophytoeukaryotes (Picoeuk), nanophytoeukaryotes (Nanoeuk) and cryptophyte-like cells (Crypto) and the second one, using a higher amplification of the photodetector of the red fluorescence signal (induced by chlorophyll), was set to focus on the small size and/or cells with low chlorophyll a fluorescence, such as Prochlorococcus (Proc) and Synechococcus (Syn). The cell concentration was determined from both Trucount beads and flow rate measurements.'*

To infer higher abundances of Prochlorococcus cell within the surface and subsurface, the use of in situ hybridization with 16S rRNA-targeted oligonucleotides would help, but unfortunately this type of measurements is not available for this cruise. Pigments data are available, and the estimate of dv-chla within the surface layers during PERLE1 cruise (in autumn) were under the limits of detection of the method after filtration of 2.8 L of water before pigment extraction, whereas abundances of *Prochlorococcus* determined by cytometry ranged 600 to 3500 cells per ml. Thus, the abundance data within surface waters un autumn is

supported by the lack of observed dv-chla. When dv-chla concentration was above detection limits, we plotted the relationship between dv-chla and abundances, the linear regression was significant ($r2=0.69$, $p < 0.05$) and the slope corresponded to 1.53 fg dvchla per cell, i.e. in the range of values cited by Garzareck et al. (2007) for High Light ecotypes and Low Light ecotypes (0.93 and 1.84 fg dvChla per cell) in the Mediterranean Sea.

In the discussion section, we added a sentence dealing about the problem of dim fluorescence of Prochlorococcus cells lines 544-570 as: *'Further, we probably could not determine accurately the abundances of Prochlorococcus cells by flow cytometry despite the special setting of the machine used to specifically enhance the detection of this population having very dim fluorescence in surface, particularly in autumn, when dv-chla was above the limits of detection in the mixed layer. This is a very common feature already described in the literature (Mella-Flores et al., 2011; Reich et al., 2022).*

Note that we added also the scientists who analyzed cells by flow cytometry in the author's list: Morgane Didry and Christophe Salmeron.

References not added in the ms

Andrade, J.M. and Estévez-Pérez, M.G. (2014). Statistical comparison of the slopes of two regression lines: A tutorial. Analytica Chimica Acta, 838:1-12, https://doi.org./10.1016/j.aca.2014.04.057

Partensky, F., Hess, W.R. and Vaulot, D. (2009) Prochlorococcus, a marine Photosynthetic Prokaryote of global significance. Microbiol. Mol. Biol. Rev. 63, 106-127.

Garczarek, L., Dufresne, A., Rousvoal, S., West, N.J., Mazard, S., Marie, D., Claustre, H., Raimbault, P., Post, A.F., Scanlan, D.J. and Partensky, F. (2007). High vertical and low horizontal diversity of Prochlorococcus ecotypes in the Mediterranean Sea in summer. Microbial Ecology, 189-206.

New references added in the ms

Horii, S., Takahashi, K., Shiozaki, T., Takeda, S., Sato, M., Yamaguchi, T., Takine, S., Hashihama, F., Kondo, Y., Takemura, T., and Furuya, K. (2023). East-West Variabilities of $N_2$ Fixation Activity in the Subtropical North Pacific Ocean in Summer: Potential Field Evidence of the Phosphorus and Iron Co-Limitation in the Western Area, JGR Oceans, 128, e2022JC019249, https://doi.org/10.1029/2022JC019249

Mella-Flores, D., Mazard, S., Humily, F., Partensky, F., Mahé, F., Bariat, L., Courties, C., Marie, D., Ras, J., Mauriac, R., Jeanthon, C., Mahdi Bendif, E., Ostrowski, M., Scanlan, D.J., and Garczarek, L. (2011). Is the distribution of Prochlorococcus and Synechococcus ecotypes in the Mediterranean Sea affected by global warming? Biogeosciences 8, 2785–2804.

Moutin, T., Karl, D. M., Duhamel, S., Rimmelin, P., Raimbault, P., Van Mooy, B. A. S., and Claustre, H. (2008). Phosphate availability and the ultimate control of new nitrogen input by nitrogen fixation in the tropical Pacific Ocean, Biogeosciences, 5, 95–109, https://doi.org/10.5194/bg-5-95-2008.

Rahav, E., Herut, B., Levi, A., Mulholland, M., and Berman-Frank, I. (2013). Springtime contribution of dinitrogen fixation to primary production across the Mediterranean Sea. Ocean Sci. 0, 489-498, https://doi.org/10.5194/os-9-489-2013.

Reich, T., Ben-Ezra, T., Belkin, N., Aharonovich, D., Roth-Rosenberg, D., Givati, S., Bialik, M., Herut, B., Berman-Frank, I., Frada, M., Krom, M., D., Lehahn, Y., Rahav, E., Sherr, D. (2022). A year in the life of the Eastern Mediterranean: Monthly dynamics of phytoplankton and bacterioplankton in an ultra-oligotrophic sea. Deep-Sea Res. I 182, article 103720, https://doi.org/10.1016/j.dsr.2022.103720.

Statham, P.J., and Hart, V. (2005). Dissolved iron in the Cretan sea (eastern Mediterranean) Limnol. Oceanogr, 50: 1142-1148.

Van Wambeke, F., Gimenez, A., Duhamel, S., Dupouy, C., Lefevre, D., Pujo-Pay, M. and Moutin, T. (2018). Dynamics and controls of heterotrophic prokaryotic production in the western tropical South Pacific Ocean: links with diazotrophic and photosynthetic activity. Biogeosciences 15, 2669 – 2689, https://doi.org/10.5194/bg-15-2669-2018.

---

## Author Comment (AC2)

**RC2**: 'Comment on egusphere-2023-2578', Anonymous Referee #2, 30 Jan 2024 reply

**General comments**

The authors investigate phosphomonoesterase (PME) versus phosphodiesterase (PDE) activities in a well-known P-depleted oligotrophic environment, the Eastern Mediterranean Sea, at two contrasted seasons. They characterize maximum hydrolysis rates (Vm) and half-saturation constants (Km) of both PME and PDE activities in relation to dissolved stocks of Phosphorus : DIP, DOP and the enzymatically hydrolysable fraction of DOP. Although phosphomonoesterase, also known as alkaline phosphatase, activities have been extensively studied in P deplete oligotrophic and coastal environments during past decades, the measurements of both PME and PDE have been achieved only recently. The authors have paid particular attention on the methodology for the measurement of these activities. The results of this paper confirm the results found elsewhere that PDE activities (Vm) could be in the same order of magnitude than PME activities. PDE seem to be regulated as PME, by the availability of DIP. However the regulation of PDE by NOx:DIP ratio is also discussed as well as the occurrence of different microbial communities having different PDE expression pattern. This paper has a significant contribution to the understanding of the Phosphorus fluxes through the microbial food web, participating to the biogeochemical cycle of Phosphorus. Obviously, it is within the scope of EGUsphere.

**Response:** We appreciate that the reviewer has found the work interesting and worthwhile to consider for publication. We truly thank him/her for providing detailed and useful comments, and we have addressed carefully him/her feedback.

Specific responses to the reviewers' comments are provided below in blue, modified sentences included in the revised version in blue + italics and line numbers referring to the new revised version with track changes are highlighted in yellow.

**Specific comments**

The scientific approach and applied methods to the studies of PME and PDE activities in relation to the dissolved phosphorus pools are particularly well adapted. The measurements of nanomolar concentrations of DIP and labile DOP in such oligotrophic environments as the Mediterranean Sea are achieved with the LWCC method having a very low detection limit (1 nM). The measurements of kinetic parameters of enzymatic activities need a particular attention since methodological biases can lead to misestimated Vm and Km. The most notable divergences in existing methodologies on enzymatic activity assays in natural environment is the substrate concentrations used for assays. The range of substrate concentration significantly affect kinetic parameters estimation and it is generally recommended to use a large substrate concentration range, up to 10 Km at least. A specific literature exists on this particular bias which could be cited by authors in the Discussion part. However, the authors discussed their results with published literature, considering these methodological aspects, which is scarcely made while necessary for meaningful comparison.

In the revised version of the manuscript, we now address the substrate concentration bias as suggested by the referee line 473-475:

*'.. as Km and Vm depend on the concentration of fluorogenic substrate added, with recommendations to add up to 10 times the Km value to calculate Vm appropriately (Urvoy et al., 2020). In most cases only one single substrate concentration….'*

**Details**

Line 27: Define the significance of DIP the first time it appears rather than Line 42

Indeed, DIP appeared in the abstract without having been identified. We wrote 'dissolved inorganic phosphorus' in the abstract, line 27.

Line 49, 692: Labry et al. 2016 rather than 2021

Yes sorry for the mistake, this was corrected in the text line 52 and in the reference list

Line 58: precise under optimal conditions of concentrations of what ? enzyme ?

Yes it is. The sentence was modified line 62 as : *'... under optimal condition of enzyme concentration, pH and temperature…'*

Line 120: nitrite rather than « nitrites » and use « DOP » rather than its significance

The sentence was modified line 133 as : *' Other nutrient analyses (nitrate, nitrite, DOP, DIP with the classical method) were sampled….'….)*

Line 141, 637: Djaoudi et al. 2018 rather than 2017

Yes sorry for the mistake, we corrected the reference to that of 2018a in the text, as there is another Djaoudi et al. 2018 cited in the ms which became 2018b. The reference list was corrected too (lines 89, 160, 438, 767, 773)

Line 225 : concerning winter depth of Pcline, refer to Fig. 3b,c

The sentence was modified line 268 as: *' In winter, the depth of the Pcline… showing a great variability among stations (Fig 3b, c)'*

Line 395 : a little more exhausted literature on P diesters composition would be informative

A list is cited lines 504. We moved it earlier lines 67-71 as suggested as follows*: ' In aquatic environments, typical P-diesters identified are nucleotides, nucleic acids, and phospholipids coming from microorganism's intracellular material (Karl and Bjorkman, 2015), but the methodology used to estimate the P-diester pool (using also a commercially purified phosphodiesterase enzyme (Monbet et al., 2007; Yamaguchi et al., 2019)) does not allow to determine the in-situ P-diesters chemical composition in detail'*

Line 408-412 : The difficult comparison with previous studies also comes from the different substrates used, MUF- derivates (Thomson et al. 2020, Sato et al. 2013) vs paranitrophenyl-derivatives (Huang et al. 2022), corresponding to different enzyme affinity. Conditions of incubation, particularly temperature may also differ between studies, optimal versus in situ temperature.

We fully agree with this comment. Part of the discussion is already dedicated to the problem of the different types of substrates used (lines 482-485 about Km, lines 505-513 about TT).

As suggested by the referee, we considered other difficulties encountered for literature comparison adding a new sentence at the end of this paragraph (lines 477-482) as: *'In addition, while some authors used MUF-derivates (Sato et al., 2013; Thomson et al., 2020), others used paranitrophenyl- derivatives (Huang et al., 2022), corresponding probably to different enzyme affinity. In addition, conditions of incubation may differ, some authors using in situ or close-to in situ temperature (Sato et al., 2013; Suzumura et al., 2012; Yamaguchi et al., 2019; Thomson et al., 2020) and others optimal temperatures (Huang et al., 2022).'*

Line 425 : Thomson et al., 2020 rather than 2019

Yes sorry for the mistake, we have corrected this reference line 498

Line 478 – 483 : discussion on Km PME >> LDOP : do the authors mean that enzymes experience locally higher substrate concentrations due to intermittent and patchy distribution of organic Phosphorus ? Could the authors explain it more precisely

The sentence was modified lines 589-602 as: *'Possibly intermittent sources and patchiness of $L_{DOP}$ composition and concentration could explain high Km relative to $L_{DOP}$ so that microorganisms maximize their PME activities at high $L_{DOP}$ concentrations. Patchiness is the consequence of the organic matter continuum of size with different molecular composition from low molecular weight to high molecular weight (Young and Ingall, 2010). Patchiness is provoked for instance, during the passage of sedimenting particles with their associated plumes (Kiørbe et al., 2001, phases of intense lysis of cells, egestion of food vacuoles by grazers (Nagata and Kirchman, 1992), or hydrolysis of particulate detritus. In addition, since most PME comes from intracellular or periplasm of cells (Luo et al., 2009), they are probably adapted to higher concentrations of DOP than that estimated by the bulk DOP measurement.'*

Figure 7 : The frame around the legend on the Km versus DIP graph could be removed

This has been done, as well as for Fig S2.

New references added in the ms:

Karl, D.M., and Björkman, K.M.: Dynamics of Dissolved Organic Phosphorus. In: Hansell D.A., and Carlson, C.A. (eds.) Biogeochemistry of Marine Dissolved Organic Matter, pp. 233-334. Burlington: Academic Press, 2015

Kiørboe, T., Ploug, H., and Thygesen, U.H.: Fluid motion and solute distribution around sinking aggregates. I. Small-scale fluxes and heterogeneity of nutrients in the pelagic environment. Mar. Ecol. Prog. Ser 211, 1-13, 2001.

Nagata, T., and Kirchman, D.L.: Release of dissolved organic matter by heterotrophic protozoa: implications for the microbial food webs. Arch. Hydrobiol. Beih. Ergebn. Limnol. 35, 99-109, 1992.

Urvoy, M., Labry, C., Delmas, D., Layla, C., and L'Helguen, S.: Microbial enzymatic assays in environmental water samples: impact of Inner Filter Effect and substrate

concentrations. Limnology and Oceanography: Methods 18, 725-738. https://doi.org/10.1002/lom3.10398, 2020.

Young, C.L., and Ingall, E.D.: Marine Dissolved Organic Phosphorus Composition: Insights from Samples Recovered Using Combined Electrodialysis/Reverse Osmosis. Aquat Geochem 16, 563-574. https://doi.org/10.1007/s10498-009-9087-y, 2010.